# Principal fitted component framework for robust support vector regression based on bounded loss: A simulation study with potential applications

**Aiman Tahir**[ID]*, **Maryam Ilyas**[¤]

College of Statistical Sciences, University of the Punjab, Lahore, Pakistan

¤ Current address: University of Liverpool, United Kingdom
* 3520254980368@pu.edu.pk

## Abstract

The inferential results regarding estimates of Support Vector Regression (SVR) are highly influenced by anomalies and ill-conditioned predictors. Excessive dimensions of data also make the model complex. To improve estimation accuracy, this paper introduces two modelling frameworks, Principal Component Robust Support Vector Regression (PCRSVR) and Principal Fitted Component Robust Support Vector Regression (PFCRSVR). These techniques are developed by incorporating PCs and PFCs with Exponential Quantile SVR (EQSVR), which is capable of dealing with ill-conditioned regressors, extreme observations, and high-dimensional data settings simultaneously. An extensive simulation study has been conducted to evaluate the performance of the proposed methods. Different evaluation criteria are chosen in this regard. Additionally, real-life data applications illustrate the efficacy of the proposed techniques as compared to competing ones.

## 1. Introduction

Principal Component Regression (PCR) [1,2] is a widely used technique to address the problem of multicollinearity within the framework of multiple linear regression. PCR is conducted in two main steps. First, Principal Component Analysis (PCA) [3,4] is performed to transform the original predictors into a new set of orthogonal components, or Principal Components (PCs). Then, a subset of these PCs is selected as new explanatory variables in the regression model.

While various adaptations of PCR are discussed in the literature (e.g., [5–7]), this study focuses on the classical PCR approach. In classical PCR, PCs with large eigenvalues are prioritized to capture the maximum variation of data. However, this approach may not always be ideal for predictive accuracy, as PCs with smaller eigenvalues could have a stronger correlation with the response variable (e.g., [8–9]). To address this limitation, various strategies have been developed to incorporate

**Data availability statement:** The data is made available at https://github.com/aiman-4/PCRSVR_PFCRSVR.git.

**Funding:** The author(s) received no specific funding for this work.

**Competing interests:** The authors have declared that no competing interests exist.

information from the response variable during the construction of PCs (e.g., [10–12]). The focus of this paper is Principal Fitted Component Regression (PFCR), proposed by [10]. It modifies PCR by regressing the response variable on a subset of Principal Fitted Components (PFCs) rather than on traditional PCs. These PFCs are designed to retain the predictive information about the response variable that is embedded within the predictors. This is often done employing inverse regression. Moreover, PFCR addresses the effects of ill-conditioned predictors [9].

Despite these advancements, PCR remains sensitive to outliers, which can distort both the PCA and the regression model. To make PCR more robust, researchers have developed estimators that combine outlier-resistant techniques with PCR. For instance, [13] proposed a robust approach to PCR that substitutes classical PCA with robust PCA. In this method, the covariance matrix is estimated using the least median of squares [14], which reduces the influence of outliers. Additional robust PCR methods have been developed to address different data complexities. [15], for example, introduced an outlier detection method for the response matrix. This method, called "resampling by halfmeans" [16], identifies and removes outlier-contaminated samples before conducting PCA. [17] proposed a robust PCR approach based on projection pursuit [18]. This approach identifies robust PCs and uses them in least-trimmed square regression [19] to reduce the influence of extreme values. [20] developed two variations of robust PCR, each tailored to different data dimensions. For low-dimensional data ($p < n$), they used the minimum covariance determinant estimator [14] to estimate the covariance matrix. For high-dimensional data ($p > n$), they recommended the ROBPCA method [21]. This method computes robust PCs specifically for high-dimensional scenarios and then applies robust regression. In addition to these methods, [22] proposed an empirical technique for robust PCR that depends upon "principal sensitive vectors" [23]. It detects outliers before performing classical PCR. [24] conducted a comparative study between robust PCR and robust partial least squares regression. Their study evaluates these methods based on efficiency, robustness, predictive competency, and model fitness. More recent techniques have incorporated advanced statistical frameworks to increase the robustness of PCR. [25] proposed an estimator for parameter function in functional logistic regression to handle functional outliers. [26] introduced a Bayesian approach to improve outlier resistance for both independent and dependent factors. This method penalizes unusual data points to certify that predictions align with the core data distribution. [27] further advanced robust PCR techniques by proposing a correlation scaled robust estimator for PCR. This method addresses the challenges of multicollinearity, outliers, and high-dimensional data. It incorporates response variable information directly into the computation of PCs. This approach enhances the predictive stability of PCR while controlling for data irregularities and dimensionality issues in multiple linear regression.

Support Vector Regression (SVR) was introduced by [28] as a method for tackling regression problems in machine learning. It depends upon the principles of Support Vector Machines (SVM) [29]. Unlike conventional regression models, SVR has garnered widespread attention across numerous disciplines [30]. The primary concern

of SVR is to minimize the deviation between the predicted outcome and actual value. Several loss functions are utilized to quantify this distance. Although classical SVR has gained notable achievement in various fields it does not encounter challenges against outliers because of utilizing unbounded loss functions. These unbounded loss functions cause the infinite increase in loss term as error increases. Consequently, a significant shift in the regression line is occurred which reduced model accuracy. To counter this issue, experts have put attention to integrating bounded loss functions into the framework of SVR. For instance, [31] introduced a truncated ε-insensitive loss to develop a truncated SVR model, motivated by the Ramp loss. [32] introduced the RLS-SVR model by truncating the least squares loss. Similarly, [33] proposed the RLNPSVR model by applying Ramp-type loss in nonparallel SVR. [34] proposed the NQSVR model depending upon a non-convex quadratic ε-insensitive loss. Nevertheless, the truncation of loss functions introduces non-differentiable points, which increases the complexity of the optimization process. [35] addressed this issue by applying the Rhinge loss in Twin Support Vector Regression (TSVR), resulting in a more robust TSVR model. More recently, a novel bounded framework is proposed by [36]. It transforms unbounded loss functions into bounded ones, which establishes the foundation for the development of BLSSVR. Inspired by these advancements, EQSVM and EQSVR are proposed by [37], based on bounded exponential quantile loss. This framework offers an alternative approach to scaling unbounded convex loss functions, providing greater resistance to outliers while preserving model efficiency.

The literature validates that blended estimators can outperform single estimators by combining the strengths of each [38]. Examples of such blended approaches include the combined *r-k* estimator, which integrates the PCA and ridge estimator [38], the robust-stein estimator [39], the combined PC-KL estimator [40], and the hybrid PC-SVR [41].

Modern data analysis is increasingly characterized by complex challenges, including multicollinearity, excessive dimensions of data, and the pervasive presence of anomalies. Traditional regression techniques often fail to deliver reliable results in such scenarios, leaving a critical gap in the ability to model real-world data effectively. This study addresses these pressing issues by introducing two approaches, i.e., PCRSVR and PFCRSVR. These methods integrate PCs and PFCs with Exponential Quantile Support Vector Regression (EQSVR) within a machine learning framework. The proposed techniques are designed to handle ill-conditioned regressors, anomalies, and large dimensions of data simultaneously. Their computational algorithms are also developed. Notably, PFCRSVR addresses the predictive limitations identified by [8] and [9] by incorporating response variable information directly into the computation of PCs. This approach aims to improve predictive accuracy by retaining components that are more relevant to the response variable. A comparative analysis is conducted among the proposed robust approaches and their non-robust counterparts to evaluate the effectiveness of the proposed techniques. Among the proposed methods and baseline counterparts, PFCRSVR consistently performs best, achieving the lowest MSE and MAE across all techniques. This establishes PFCRSVR as the most effective framework for complex data environments.

The organization of the paper is as follows: Subsections of section 1 describe the principal component regression, principal fitted component regression and robust support vector regression, respectively. The proposed methodology and its computational algorithms are discussed in section 2. Section 3 conducts the simulation study to investigate the performance of the proposed methods. Real-life data applications illustrate the developed techniques in section 4. Section 5 gives concluding remarks on the paper.

## 1.1. Principal component regression

[42] introduced Principal Component Analysis (PCA) as a method to transform correlated predictors into uncorrelated variables called principal components (PCs). Each PC is a combination of the original predictors, constructed using specific weights. Consider $\boldsymbol{X}$, an $n \times p$ matrix where $n$ is the number of observations and $p$ is the number of predictors. The PCs are computed such as $\boldsymbol{m}_1 = \boldsymbol{v}_1^T \boldsymbol{x}_1, \boldsymbol{m}_2 = \boldsymbol{v}_2^T \boldsymbol{x}_2, \ldots, \boldsymbol{m}_p = \boldsymbol{v}_p^T \boldsymbol{x}_p$. Here, $\boldsymbol{v}_1, \boldsymbol{v}_2, \ldots, \boldsymbol{v}_p$ represent eigenvectors of the covariance matrix ($\boldsymbol{\Sigma} = cov(\boldsymbol{X})$), and their corresponding eigenvalues are $\lambda_1, \lambda_2, \ldots, \lambda_p$. The eigenvectors ($\boldsymbol{v}_1, \boldsymbol{v}_2, \ldots, \boldsymbol{v}_p$) are arranged into $p \times p$ matrix ($\boldsymbol{V}$) and PCs ($\boldsymbol{m}_1, \boldsymbol{m}_2, \ldots, \boldsymbol{m}_p$) are composed into $n \times p$ matrix ($\boldsymbol{M}$).

In PCR, a subset of the first $q$-PCs ($M_q$) is used to model the response variable ($y$) with $q \leq p$. This relationship is modelled by Eq 1, here $\alpha$ is the q×1 vector of regression coefficients for the q-PCs, and $\epsilon$ is the (n×1) error term. The regression coefficients ($\alpha_q$) are estimated using the least squares method (Eq 2). Once these coefficients are estimated, they are transformed back to the original predictor space, as defined in Eq 3. Here, $\hat{\beta}$ represents the (p×1) vector of estimated regression coefficients regarding original predictors. By selecting only the leading PCs that account for most of the variability in the data, PCR simplifies the model and addresses the problem of ill-conditioned regressors.

$$y = M_q\alpha_q + \epsilon \tag{1}$$

$$\hat{\alpha}_q = (M_q^T M_q)^{-1} M_q^T y \tag{2}$$

$$\hat{\beta} = V_{p \times q}\hat{\alpha}_q \tag{3}$$

## 1.2. Principal fitted component regression

Using principal components as regressors in regression models raises certain concerns. First, PCs are derived solely from the predictor variables without incorporating the response variable. This approach assumes that the response depends primarily on the first few PCs, but in reality, it might also rely on components associated with smaller variations. Second, PCs lack the properties of invariance and equivariance when the predictor variables undergo full-rank linear transformations.

To address these limitations, PFCs were introduced for dimension reduction in regression modeling [10]. Compared to PCs, PFCs provide two key advantages. They retain equivariance under full-rank linear transformations of predictors and can be tailored to incorporate information from the response variable.

PFCs are constructed by extracting sufficient information about the response variable ($y$) from the predictors ($X$). This is often achieved through inverse regression, which involves estimating E[X | y = y]. Unlike forward regression, which models E[y | X = x], inverse regression reduces the problem to $p$ times one-dimensional regressions.

The Eq 4 is an inverse regression model such that $X_y \sim N(\mu + \Gamma v_y, \Delta)$. Here, $\mu = E(X)$ represents the mean of the predictors, and $\Gamma \in R^{p \times q}$ is a semi-orthogonal matrix whose columns form a basis for the q-dimensional subspace $S_\Gamma = span\{\mu_y - \overline{\mu}|y\epsilon S_y\}$, where $S_y$ is the sample space of y. The term $v_y = \Upsilon f_y$ includes $f_y \in R^r$ and $\Upsilon \in R^{q \times r}$ with $q \leq \min (r, p)$, a mean-centred vector-valued function of y, satisfying $\Sigma_y f_y = 0$. Instead of indexing predictors conventionally by $i$, here y serves as the index. The predictors ($X_y$) are regressed on a response-dependent function ($f_y$), which is constructed using a specific basis function g. This basis is mean-centred as $f_y = g_y - \overline{g}$, with $g_y$ typically chosen as a polynomial basis with degree $r$, i.e., $g_y = (y, y^2, \ldots, y^r)^T$ and $\varepsilon \sim N(0, \Delta)$. Here, $\Delta$ assumes independence of $y$ and its simplest form is isotropic with $\Delta = \sigma^2 I_p$.

$$X_y = \mu + \Gamma v_y + \varepsilon \tag{4}$$

To compute PFCs, the sample covariance matrix of the fitted predictors, $\widehat{\Sigma}_{fit} = \frac{\hat{X}^T \hat{X}}{n}$, is estimated. Here, $\hat{X}$ represents the predictors fitted from the regression of $X_y$ on $f_y$.

PCA is then applied to $\hat{\Sigma}_{fit}$, yielding eigenvectors $\hat{\Phi}_1^T, \hat{\Phi}_2^T, \ldots, \hat{\Phi}_p^T$ corresponding to eigenvalues $\hat{\lambda}_1, \hat{\lambda}_2, \ldots, \hat{\lambda}_p$. These eigenvectors are used to construct PFCs, expressed as $\hat{\Phi}_1^T x_1, \hat{\Phi}_2^T x_2, \ldots, \hat{\Phi}_k^T x_p$. Instead of using all PFCs, a subset of $q$-PFCs is employed in the regression model. Since PFCs incorporate information from the response variable during their construction, they often outperform PCs in regression tasks under various scenarios [10].

### 1.3. Exponential Quantile Support Vector Regression (EQSVR)

Consider, we have $n$ training instances and $p$ features. The $i^{th}$ training instance can be denoted as $\boldsymbol{x}_i \in R$ and its associated outcome can be denoted as $y_i$, $i = 1, 2, ...., n$. The data matrix ($\boldsymbol{X} \in R^{n \times p}$) can be composed by arranging samples in rows and features in columns and $\boldsymbol{y}$ is the ($n \times 1$) vector of responses. [37] introduced two parameters of exponential quantile loss ($L_{eq}(u) = \eta(\frac{1}{1+\exp(-\lambda l_{pin}(u)+10\tau)} - \phi)$) in standard SVR. Here, $\lambda > 0$ and $\tau \geq 0$ are two tuning parameters. $\lambda$ controls the steepness of $L_{eq}$-loss and $\tau$ acts as a hedging factor. $l_{pin}(u)$ denotes Pinball loss and $\varphi = \frac{1}{1+\exp(10\tau)}$ represents location constant satisfying $L(0) = 0$. Also, $\eta = \frac{1+\exp(10\tau)}{\exp(10\tau)}$ denotes the normalizing constant ensuring $L(\infty) = 1$. Thus, the objective function of EQSVR is formulated in Eq 5. Here, $\boldsymbol{w}$ is the ($p \times 1$) vector of weights, $b$ denotes bias and $C$ represents the non-negative penalty parameter. After estimating $\boldsymbol{w}$ and $b$ we can predict a new sample $\boldsymbol{x}_{new}$ by using relation $f(\boldsymbol{x}_{new}) = \boldsymbol{w}^T \boldsymbol{x}_i + b$.

$$\min_{\boldsymbol{w}} \frac{1}{2}(\| \boldsymbol{w} \|_2^2 + b^2) + C\sum_{i=1}^{n} L_{eq}(y_i - (\boldsymbol{w}^T\boldsymbol{x}_i + b)) \tag{5}$$

In this paper, EQSVR is formulated for a linear regression problem. Let's assume $\check{\boldsymbol{w}} = (\boldsymbol{w}^T, b)^T$ and $\check{\boldsymbol{X}} = (\boldsymbol{X}, \boldsymbol{e})$. Here, $\boldsymbol{e}$ is the ($n \times 1$) vector of ones. According to these notations, the objective function (Eq 5) is transformed to Eq 6.

$$\min_{\check{\boldsymbol{w}}} \frac{1}{2}\| \check{\boldsymbol{w}} \|_2^2 + C\sum_{i=1}^{n} L_{eq}(y_i - \check{\boldsymbol{w}}^T\check{\boldsymbol{x}}_i) \tag{6}$$

EQSVR utilizes the ConCave-Convex Procedure (CCCP) to transform non-convex $L_{eq}$-loss into the chain of convex optimization problems. Then, these convex optimization problems are solved by ClipDCD algorithm [43]. To solve Eq 6, $L_{eq}$-loss is decomposed into $g(u)$ and $h(u)$ defined in Eq 7 and Eq 8, respectively. Subsequently, the model of EQSVR is formulated in Eq 9.

$$g(u) = \left( \frac{\lambda^2 \eta l_{pin}(u)}{10} + \frac{\lambda}{1+\exp(10\tau)} \right) l_{pin}(u) \tag{7}$$

$$h(u) = -\left( \frac{\lambda^2 \eta l_{pin}(u)}{10} + \frac{\lambda}{1+\exp(10\tau)} \right) l_{pin}(u) + \eta \left( \frac{1}{1+\exp(-\lambda l_{pin}(u)+10\tau)} - \varphi \right) \tag{8}$$

$$\min_{\check{\boldsymbol{w}}} \frac{1}{2}\| \check{\boldsymbol{w}} \|_2^2 + C\sum_{i=1}^{n} g(y_i - \check{\boldsymbol{w}}^T\check{\boldsymbol{x}}_i) + C\sum_{i=1}^{n} h(y_i - \check{\boldsymbol{w}}^T\check{\boldsymbol{x}}_i) \tag{9}$$

The first two terms of Eq 9 are convex parts and are jointly represented by $L_{vex}(\check{\boldsymbol{w}})$. The third term is the concave part and is denoted by $L_{cav}(\check{\boldsymbol{w}})$. The CCCP method is employed to optimize the problem defined in Eq 9. The subsequent sub-problems (Eq 10) are addressed to iteratively obtain the optimal solution. Here, $\nabla L_{cav}(\check{\boldsymbol{w}}^k)$ is the derivative of $L_{cav}(\check{\boldsymbol{w}}^k)$ for obtaining optimal solution $\check{\boldsymbol{w}}^k$. An auxiliary variable ($\delta^k = (\delta_1^k, \delta_2^k, ...., \delta_n^k)^T$) defined in Eq 11 is introduced for ease of notation. Then, the Eq 9 is reformulated to the Eq 12 and is further simplified to Eq 13. Here, $\xi_i = \lambda l_{pin} \left( y_i - \check{\boldsymbol{w}}^T\check{\boldsymbol{x}}_i \right) = \lambda max \left( y_i - \check{\boldsymbol{w}}^T\check{\boldsymbol{x}}_i, -\tau(y_i - \check{\boldsymbol{w}}^T\check{\boldsymbol{x}}_i) \right)$ and Eq 14 is a matrix form of Eq 13.

$$\check{\boldsymbol{w}}^{k+1} = arg\min_{\boldsymbol{w}} L_{vex}(\check{\boldsymbol{w}}) + \nabla L_{cav}(\check{\boldsymbol{w}}^k)^T \check{\boldsymbol{w}} \tag{10}$$

$$\delta_i^k = \begin{cases} \dfrac{2\lambda^2(y_i - \check{\boldsymbol{w}}^T\check{\boldsymbol{x}}_i)}{5} + \dfrac{\lambda}{2} - \dfrac{2\lambda exp(-\lambda(y_i - \check{\boldsymbol{w}}^T\check{\boldsymbol{x}}_i))}{\left(1 + \exp\left(-\lambda(y_i - \check{\boldsymbol{w}}^T\check{\boldsymbol{x}}_i)\right)\right)^2}, & y_i - \check{\boldsymbol{w}}^T\check{\boldsymbol{x}}_i \geq 0 \\ \dfrac{2\lambda^2\tau^2(y_i - \check{\boldsymbol{w}}^T\check{\boldsymbol{x}}_i)}{5} + \dfrac{\lambda\tau}{2} - \dfrac{2\lambda\tau exp(-\lambda\tau(y_i - \check{\boldsymbol{w}}^T\check{\boldsymbol{x}}_i))}{\left(1 + \exp\left(-\lambda\tau(y_i - \check{\boldsymbol{w}}^T\check{\boldsymbol{x}}_i)\right)\right)^2}, & 1 - y_i\check{\boldsymbol{w}}^T\check{\boldsymbol{x}}_i < 0 \end{cases} \tag{11}$$

$$\min_{\check{\boldsymbol{w}}} \frac{1}{2} \| \check{\boldsymbol{w}} \|_2^2 + C \sum_{i=1}^n g(y_i - \check{\boldsymbol{w}}^T\check{\boldsymbol{x}}_i) + C \sum_{i=1}^n \delta_i^k \check{\boldsymbol{x}}_i^T \check{\boldsymbol{w}} \tag{12}$$

$$\min_{\check{\boldsymbol{w}}} \frac{1}{2} \| \check{\boldsymbol{w}} \|_2^2 + C \sum_{i=1}^n \left( \frac{\lambda^2 l_{pin}^2(y_i - \check{\boldsymbol{w}}^T\check{\boldsymbol{x}}_i)}{5} + \frac{\lambda l_{pin}(y_i - \check{\boldsymbol{w}}^T\check{\boldsymbol{x}}_i)}{2} \right) + C \sum_{i=1}^n \delta_i^k \check{\boldsymbol{x}}_i^T \check{\boldsymbol{w}}$$

$$\min_{\check{\boldsymbol{w}}} \frac{1}{2} \check{\boldsymbol{w}}^T \check{\boldsymbol{w}} + C \sum_{i=1}^n \left( \frac{\xi_i^2}{5} + \frac{\xi_i}{5} \right) + C \sum_{i=1}^n \delta_i^k \check{\boldsymbol{x}}_i^T \check{\boldsymbol{w}} \tag{13}$$

$$\min_{\check{\boldsymbol{w}}, \boldsymbol{\xi}} \frac{1}{2} \check{\boldsymbol{w}}^T \check{\boldsymbol{w}} + \frac{1}{5} C \boldsymbol{\xi}^T \boldsymbol{\xi} + \frac{C}{2} \boldsymbol{e}^T \boldsymbol{\xi} + C \delta^{kT} \check{\boldsymbol{X}} \check{\boldsymbol{w}} \text{ Subject to } \begin{cases} \lambda \left( \boldsymbol{y} - \check{\boldsymbol{X}}\check{\boldsymbol{w}} \right) \leq \boldsymbol{\xi} \\ \tau\lambda \left( \boldsymbol{y} - \check{\boldsymbol{X}}\check{\boldsymbol{w}} \right) \leq \boldsymbol{\xi} \end{cases} \tag{14}$$

The Lagrange function is defined in Eq 15 by incorporating two variables γ and θ. The Karush-Kuhn-Tucker (KKT) conditions are derived in Eqs 16–19 and must be satisfied. The resulting Eq 20 is obtained by plugging the KKT conditions in Lagrangian function (Eq 15). After solving Eq 16, we get the weight vector ($\check{\boldsymbol{w}}$) that is defined in Eq 25. Hence, Eq 23 can be redefined after utilizing the results mentioned in Eq 21 and Eq 22. Here, $I$ denote the identity matrix and $\boldsymbol{0}$ is the vector of zeroes.

$$L \left( \check{\boldsymbol{w}}, \boldsymbol{\xi}, \gamma \right) = \frac{1}{2} \check{\boldsymbol{w}}^T \check{\boldsymbol{w}} + \frac{1}{5} C \boldsymbol{\xi}^T \boldsymbol{\xi} + \frac{C}{2} \boldsymbol{e}^T \boldsymbol{\xi} + C \delta^{kT} \check{\boldsymbol{X}} \check{\boldsymbol{w}} + \gamma^T \left( \lambda \left( \boldsymbol{y} - \check{\boldsymbol{X}}\check{\boldsymbol{w}} \right) - \boldsymbol{\xi} \right) + \theta^T(\tau\lambda \left( \boldsymbol{y} - \check{\boldsymbol{X}}\check{\boldsymbol{w}} \right) - \boldsymbol{\xi}) \tag{15}$$

$$\frac{\partial L}{\partial \check{\boldsymbol{w}}} = \check{\boldsymbol{w}} + \check{\boldsymbol{X}}^T \left( C\delta^k - \lambda\gamma + \tau\lambda\theta \right) = 0; \tag{16}$$

$$\frac{\partial L}{\partial \boldsymbol{\xi}} = \frac{2C\boldsymbol{\xi}}{5} + \frac{C\boldsymbol{e}}{2} - \gamma - \theta = 0; \tag{17}$$

$$\frac{\partial L}{\partial \gamma} = \lambda \left( \boldsymbol{y} - \check{\boldsymbol{X}}\check{\boldsymbol{w}} \right) - \boldsymbol{\xi} = 0; \tag{18}$$

$$\frac{\partial L}{\partial \boldsymbol{\beta}} = \tau\lambda \left( \check{\boldsymbol{X}}\check{\boldsymbol{w}} - \boldsymbol{y} \right) - \boldsymbol{\xi} = 0 \tag{19}$$

$$\max_{\gamma, \theta} -\frac{1}{2} \left( \lambda\gamma - \tau\lambda\theta - C\delta^k \right)^T \check{\boldsymbol{X}}\check{\boldsymbol{X}}^T \left( \lambda\gamma - \tau\lambda\theta - C\delta^k \right) + \boldsymbol{y}^T \left( \lambda\gamma - \tau\lambda\theta \right)$$

$$- \frac{5}{4C} \left( \gamma + \theta - \frac{C\boldsymbol{e}}{2} \right)^T \left( \gamma + \theta - \frac{C\boldsymbol{e}}{2} \right) \text{ subject to } \gamma \geq 0, \ \theta \geq 0 \tag{20}$$

$$\boldsymbol{K}_r = \begin{pmatrix} \lambda^2 \check{\boldsymbol{X}}\check{\boldsymbol{X}}^T & -\tau\lambda^2 \check{\boldsymbol{X}}\check{\boldsymbol{X}}^T \\ -\tau\lambda^2 \check{\boldsymbol{X}}\check{\boldsymbol{X}}^T & \tau^2\lambda^2 \check{\boldsymbol{X}}\check{\boldsymbol{X}}^T \end{pmatrix}, \ \boldsymbol{I}_r = \begin{pmatrix} \lambda\boldsymbol{y} \\ -\lambda\tau\boldsymbol{y} \end{pmatrix} + \frac{5}{2} \begin{pmatrix} \frac{\boldsymbol{e}}{2} - \frac{\delta^k}{\lambda} \\ \frac{\boldsymbol{e}}{2} - \frac{\delta^k}{\lambda} \end{pmatrix} \tag{21}$$

$$u = \begin{pmatrix} \gamma - \frac{C\delta^k}{\lambda} \\ \theta \end{pmatrix}, \ R = \begin{pmatrix} I & I \\ I & I \end{pmatrix}, \ lb = \begin{pmatrix} -\frac{C\delta^k}{\lambda} \\ 0 \end{pmatrix} \tag{22}$$

$$\min_{u} \frac{1}{2} u^T \left( K_r + \frac{5}{2C} R \right) u - l_r^T u \text{ subject to } u \geq lb \tag{23}$$

The problem in Eq 23 is found a quadratic optimization problem that can be solved by the ClipDCD algorithm [43]. The values of $\delta^k$ and $u^k$ are iteratively updated over the CCCP iteration. After obtaining $u^*$, we can predict for new instance by following the Eq 24. Here, $u_i^*$ denotes the optimal solution. The CCCP algorithm depends upon ClipDCD EQSVR which is described in Fig 1.

$$f(\boldsymbol{x}) = \sum_{i=1}^{n} u_i^* \check{\boldsymbol{x}}_i \tag{24}$$

$$\check{w} = \check{X}^T (\lambda\gamma - \tau\lambda\theta - C\delta^k) \tag{25}$$

## 2. Proposed methods

### 2.1. Principal Component Robust Support Vector Regression (PCRSVR)

The proposed PCRSVR is a hybrid technique that combines the PCs and EQSVR in a machine learning framework. This approach can handle the data irregularities, ill-conditioned predictors and excessive data dimensions simultaneously. It first performs PCA on predictors and constructs new transformed variables known as principal components, eliminating the problem of ill-conditioned predictors. It chooses the first $q$-PCs that explain the maximum variation of the predictors. Then, these PCs are used as regressors in the EQSVR framework to model outcome variable that is characterized by anomalies. These anomalies are tackled with $L_{eq}$-loss that is plugged into EQSVR.

The PCRSVR performs the following steps to calculate the MSE of estimated regression parameters.

1. Generate predictors ($\boldsymbol{X}$) by Eq 27 and standardize them.

2. Simulate the response variable ($\boldsymbol{y}$) using Eq 28. Define the vector of regression coefficients ($\boldsymbol{\beta}$) as an eigenvector relevant to the largest eigenvalue of the information matrix ($\boldsymbol{X}^T \boldsymbol{X}$).

3. Introduce outliers in $\boldsymbol{y}$ using Eq 29 according to the outliers fraction specified in section 3.

4. Obtain eigenvalues ($\lambda_1, \lambda_2, \ldots, \lambda_p$) and eigenvectors ($\boldsymbol{V} = (\boldsymbol{v}_1, \boldsymbol{v}_2, \ldots, \boldsymbol{v}_p)$) of ($\boldsymbol{X}^T \boldsymbol{X}$) by applying PCA and construct new transformed variables ($\boldsymbol{M} = \boldsymbol{XV}$).

5. Retain $q$-PCs ($\boldsymbol{M}_q$), where $q$ represents the number of components explaining at least 80% of the variation of $\boldsymbol{X}$.

6. Model contaminated $\boldsymbol{y}$ based on $\boldsymbol{M}_q$ using linear kernel and considering $L_{eq}$-loss in SVR described in subsection 1.3.

7. Estimate regression parameters for $q$-PCs using Eq 25 based on the modelling framework implemented in step 6.

8. Convert these estimated parameters back to the original predictor space using the transformation explained in Eq 3.

9. Calculate the MSE of $\hat{\beta}_{PCRSVR}$ according to Eq 26. Here, $\hat{\beta}$ denotes the estimated value through the proposed modelling framework (PCRSVR) and $\beta$ represents its respective true value.

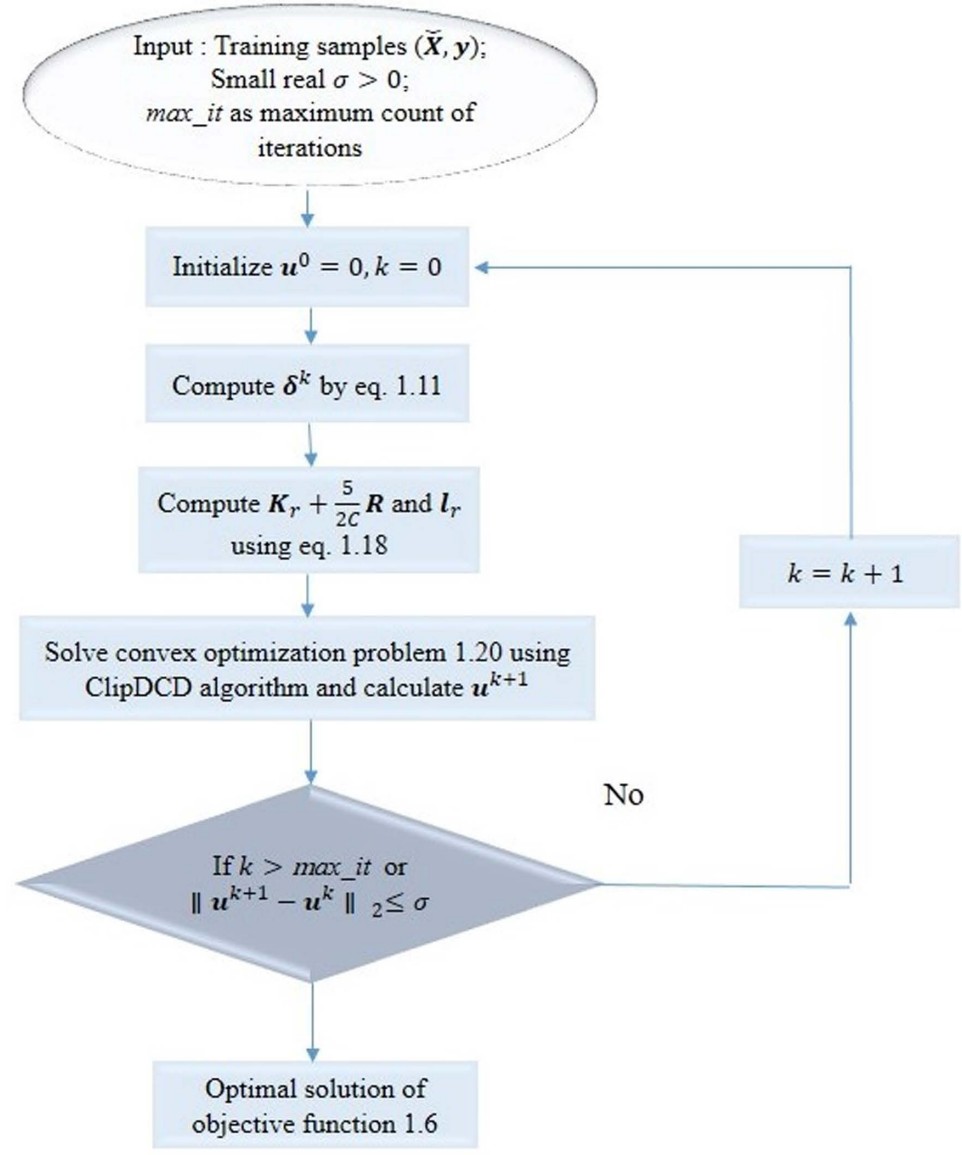

**Fig 1. The CCCP algorithm of EQSVR based on ClipDCD.**

$$MSE(\hat{\boldsymbol{\beta}}) = \frac{1}{p}\sum_{l=}^{p}\left(\hat{\beta}_l - \beta_l\right)^2$$

(26)

10.    Replicate steps 1–9 for 100 Monte Carlo runs and obtain a mean over 100 runs.

## 2.2. Principal Fitted Component Robust Support Vector Regression (PFCRSVR)

The PFCRSVR method combines PFCs with EQSVR to provide a robust solution for data irregularities, ill-conditioned predictors, and high-dimensional settings simultaneously. It also addresses the challenge noted by [8] and [9] by

incorporating fitted predictors during the computation of PCs instead of original ones. The computational process of fitted predictors is described in subsection 1.2. In PFCRSVR, PCA is applied to fitted predictors and construct PFCs as detailed in subsection 1.2. From these PFCs, the top $q$-PFCs are selected for further modeling. These components are then used as inputs in the EQSVR framework to predict the outcome variable that contains anomalies. The $L_{eq}$-loss function within EQSVR ensures resilience to these anomalies, enabling accurate and reliable regression modeling.

The following steps are involved to obtain the MSE of estimated regression parameters of PFCRSVR.

1. Perform steps (1–3) of PCRSVR's algorithm.

2. Compute fitted predictors ($\hat{\boldsymbol{X}}_{n \times p}$) by regressing $\boldsymbol{X}$ on the polynomial basis of $\boldsymbol{y}$ performing the inverse regression model described in Eq 4. Here, the PFC model assumes isotropic structure and second-degree polynomial ($r = 2$).

3. Obtain fitted sample covariance matrix ($\hat{\Sigma}_{fit}$) of fitted predictors ($\hat{\boldsymbol{X}}_{n \times p}$).

4. Perform PCA on fitted predictors ($\hat{\boldsymbol{X}}_{n \times p}$) to obtain eigenvalues ($\hat{\lambda}_1, \hat{\lambda}_2, \ldots, \hat{\lambda}_p$) and their corresponding eigenvectors ($\hat{\Phi}_1^T, \hat{\Phi}_2^T, \ldots, \hat{\Phi}_p^T$).

5. Multiply eigenvectors ($\hat{\Phi}_1^T, \hat{\Phi}_2^T, \ldots, \hat{\Phi}_p^T$) with $\boldsymbol{X}$ to get PFCs ($\hat{\Phi}_1^T \boldsymbol{x}_1, \hat{\Phi}_2^T \boldsymbol{x}_2, \ldots, \hat{\Phi}_k^T \boldsymbol{x}_p$). Compose these PFCs in $n \times p$ matrix ($\boldsymbol{Z}$).

6. Select $q$-PFCs ($\boldsymbol{Z}_q$) that account for at least 80% of the variation of $\boldsymbol{X}$ and consider them in the further modelling process.

7. Use the $L_{eq}$-loss function in SVR to model the contaminated response variable ($\boldsymbol{y}$) based on $\boldsymbol{Z}_q$ using linear kernel as detailed in subsection 1.3.

8. Estimate regression coefficients for the retained $q$-PFCs ($\boldsymbol{Z}_q$) using Eq 25.

9. Transform the estimated coefficients back to the original predictor space considering the mapping described in Eq 3.

10. Compute MSE of proposed estimator PFCRSVR ($\hat{\beta}_{PFCRSVR}$) using Eq 26.

11. Iterate the steps 1–10 for 100 Monte Carlo runs and calculate the mean over 100 replications.

## 3. Simulation study

In this section, we evaluate the performance of the proposed methods (i.e., PCRSVR and PFCRSVR) by conducting a Monte Carlo simulation study using R programming language. The relevant code and data files are deposited at https://github.com/aiman-4/PCRSVR_PFCRSVR.git. For the implementation of PFCR and classical SVR, the R packages ldr [44] and e1071 [45] are respectively utilized. The competing techniques are SVR, EQSVR, PCSVR and PFCSVR. PCSVR and PFCSVR are two hybrid approaches that utilize $q$-PCs and $q$-PFCs as regressors into classical SVR. The simulation settings are outlined in forthcoming subsection 3.1.

### 3.1. Simulation design

In this subsection, the data generation process of synthesis data sets is described. The explanatory variables are generated according to Eq 27, following the approach of [40] and [39]. In this setup, $\rho$ indicates the correlation among two explanatory variables and $D_{ij}$ represents independent pseudo-random numbers drawn from standard normal distribution. The response variable is simulated based on Eq 28. Here, $e_i \sim N(0,1)$ and regression coefficient $\beta_j$ is chosen to satisfy $\sum_{j=1}^{p} \beta_j^2 = 1$, following the [46].

$$x_{ij} = (1 - \rho^2)^{1/2} D_{ij} + \rho D_{i,p+1} , \; i = 1, 2, 3, \ldots, n \text{ and } j = 1, 2, 3, \ldots, p \tag{27}$$

$$y_i = \sum_{j=1}^{p} \beta_j x_{ij} + e_i, \quad i = 1, 2, 3, \ldots, n \tag{28}$$

The proposed techniques are evaluated by varying several key factors including sample size, degree of correlation, level of contamination, and number of predictors. We consider collinearity levels of $\rho$ = 0.8, 0.9 and 0.99. The number of explanatory variables is set to $p$ = 5, 15 and 25. Additionally, we test sample sizes of $n$ = 50, 100, 300 and 500. To evaluate the robustness of proposed techniques, we introduce different proportions of outliers, i.e., 0%, 5%, 15%, and 30%. Different combinations of these factors are considered in this study and relevant results are reported in section 4. The values of hyperparameters of EQSVR are set as $\tau$ = 0.7 and $\lambda$ = 0.5. Whereas the value of penalty parameter ($C$) of EQSVR is chosen as 0.2 for all the scenarios except the scenarios where $p$ = 25. In this case, the value of $C$ is 0.04.

This study focuses on vertical outliers, which affect only the response variable. We contaminate the response variable ($y$) randomly, following Eq 29 as suggested by [40]. Here, $b$ is the magnitude of outliers, set at a constant value of 10.

$$y_i^* = b * \max(\mathbf{y}) + y_i \tag{29}$$

### 3.2. Performance evaluation criteria

The proposed techniques are compared with their competing ones based on Mean Square Error (MSE) and Mean Absolute Error (MAE). These evaluation measures have been considered by various researchers (see, e.g., [37,39,40]). These metrics are computed using Eq 30 and Eq 31. Here, $\hat{\beta}_l$ is the $l^{th}$ estimated regression coefficient of any studied modelling framework and $\beta_l$ is its corresponding true value. The technique that produces the lowest values of MSE and MAE is considered the most effective.

$$MSE(\hat{\boldsymbol{\beta}}) = \frac{1}{p}\sum_{l=1}^{p}\left(\hat{\beta}_l - \beta_l\right)^2 MSE(\hat{\boldsymbol{\beta}}) = \frac{1}{p}\sum_{l=1}^{p}\left(\hat{\beta}_l - \beta_l\right)^2 MSE(\hat{\boldsymbol{\beta}}) = \frac{1}{p}\sum_{l=1}^{p}\left(\hat{\beta}_l - \beta_l\right)^2 MSE(\hat{\boldsymbol{\beta}}) = \frac{1}{p}\sum_{l=1}^{p}\left(\hat{\beta}_l - \beta_l\right)^2 MSE(\hat{\boldsymbol{\beta}}) = \tag{30}$$

$$MAE(\hat{\boldsymbol{\beta}}) = \frac{1}{p}\sum_{l=1}^{p}|\hat{\beta}_l - \beta_l| \tag{31}$$

Additionally, the strength of the developed techniques against their counterparts is quantified by the improved percentage reduction in MSE regarding the proposed ones. This indicator is termed PMSE and is computed using Eq 32. Here, PMSE denotes the magnitude of percentage which increases or decreases due to the MSE of proposed techniques over their competing ones. MSE* and MSE** denote the mean square error of the proposed technique and its competitor, respectively. Theoretically, the proposed techniques attain achievement if PMSE produces a positive value. The negative value of PMSE shows the inferiority of the proposed techniques over their baseline techniques.

$$PMSE = \frac{\left(MSE^{**} - MSE^*\right)}{MSE^{**}} \times 100 \tag{32}$$

### 4. Results

An extensive simulation study has been conducted by taking various above-mentioned scenarios into account. The simulation experiments are replicated 100 times. For each replication, the MSE of $\hat{\boldsymbol{\beta}}$ and MAE of $\hat{\boldsymbol{\beta}}$ are computed for proposed methods (e.g., PCRSVR and PFCRSVR) and their competitors (e.g., SVR, EQSVR, PCSVR and PECSVR). The

summary statistics (i.e., mean and Standard Error (SE)) of performance measures over 100 replications are reported in Tables (1–12). For brevity, a few tables are inserted in supporting information (see, S1-S6 Tables). It can be noticed from Tables (1–12), that the proposed techniques (PCRSVR and PFCRSVR) produce reduced MSE and MAE as compared to their baseline counterparts (SVR, EQSVR, PCSVR and PFCSVR) in almost all studied simulations settings. Also, the increasing pattern of sample size exhibits decreasing behaviour of MSE and MAE for all the studied estimators (see, Tables 1–12). All the studied techniques perform well with various degrees of correlation and different

percentages of outliers. However, the proposed techniques PCRSVR and PFCRSVR outperform EQSVR and their respective non-robust estimators PCSVR and PFCSVR. As the contamination fraction increases, the MSE and MAE increase for all the techniques. However, these metrics for proposed techniques tend to increase with less proportion over competing ones, especially when the sample size is large (See, Table 1–12). It is also noticed that the MSE and MAE of SVR, PCSVR and PFCSVR tend to increase with the increase in level of collinearity. Whereas the inverse relationship is exhibited between the degree of correlation and performance measures of EQSVR, PCRSVR and PFCRSVR. For instance, the MSE and MAE of SVR, PCSVR and PFCSVR increase with the increase in the degree of collinearity (See, Tables 1–6). It can also be noticed that the increase in predictors generally increases the MSE and MAE of SVR and decreases

the performance metrics for all other techniques except a few cases. For instance, the direct relationship is observed among the number of predictors and performance metrics of PCSVR when $\rho = 0.99$ (see, Tables 5, 6, 11 and 12).

Moreover, Figs 2–7 and S1-S3 Figs provide more clearer view by displaying the percentage reduction in MSE of PCRSVR and PFCRSVR over their competitors EQSVR, PCSVR and PFCSVR. The proposed techniques become be most efficient due to producing the reduced MSE as compared to the competing ones. It is also evident from Figs 4 and 7 and S3 Fig, that the efficiency of PCRSVR and PFCRSVR substantially improves as compared to PCSVR and PFCSVR

**Table 1. The summary statistics (mean±S.E) of MSE of regression coefficients regarding proposed and other studied estimators for $p=5$ and $\rho=0.8$.**

| Method | Sample size n | Contamination levels 0% | 5% | 15% | 30% |
|--------|------|------|------|------|------|
| SVR | 50 | 0.0599±0.0417 | 0.0714±0.0490 | 0.1080±0.0888 | 0.2904±0.3760 |
| | 100 | 0.0329±0.0238 | 0.0390±0.0246 | 0.0542±0.0374 | 0.1015 ±0.0771 |
| | 300 | 0.0106±0.0080 | 0.0135±0.0080 | 0.0159±0.0119 | 0.0321±0.0256 |
| EQSVR | 50 | 0.0173±0.0082 | 0.0176±0.0073 | 0.0197± 0.0101 | 0.0203 ±0.0120 |
| | 100 | 0.0148±0.0077 | 0.0152±0.0080 | 0.0150±0.0087 | 0.0147 ±0.0089 |
| | 300 | 0.0079±0.0057 | 0.0091±0.0054 | 0.0082±0.0050 | 0.0086 ±0.0049 |
| PCSVR | 50 | 0.0389±0.0305 | 0.0473±0.0341 | 0.0725±0.0622 | 0.1680 ±0.2298 |
| | 100 | 0.0217±0.0176 | 0.0261±0.0191 | 0.0390±0.0342 | 0.0745±0.0632 |
| | 300 | 0.0075±0.0056 | 0.0097±0.0071 | 0.0110±0.0098 | 0.0223±0.0206 |
| PCRSVR | 50 | 0.0151±0.0075 | 0.0153±0.0072 | 0.0180±0.0098 | 0.0189 ±0.0120 |
| | 100 | 0.0119±0.0067 | 0.0122±0.0077 | 0.0133 ±0.0086 | 0.0125±0.0085 |
| | 300 | 0.0057±0.0044 | 0.0071±0.0045 | 0.0058±0.0038 | 0.0062±0.0039 |
| PFCSVR | 50 | 0.0116±0.0177 | 0.0138±0.0243 | 0.0193±0.0222 | 0.0542±0.0762 |
| | 100 | 0.0079±0.0095 | 0.0075±0.0089 | 0.0130±0.0155 | 0.0232±0.0323 |
| | 300 | 0.0026±0.0034 | 0.0033±0.0036 | 0.0038±0.0044 | 0.0058±0.0069 |
| PFCRSVR | 50 | 0.0104±0.0062 | 0.0107±0.0062 | 0.0140±0.0094 | 0.0156±0.0116 |
| | 100 | 0.0071± 0.0055 | 0.0070± 0.0057 | 0.0081±0.0067 | 0.0075±0.0057 |
| | 300 | 0.0026±0.0028 | 0.0029± 0.0028 | 0.0024±0.0022 | 0.0020±0.0019 |

**Table 2. The summary statistics (mean±S.E) of MAE of regression coefficients regarding proposed and other studied estimators for $p = 5$ and $\rho = 0.8$.**

| Method | Sample size n | 0% | Contamination Levels 5% | 15% | 30% |
|---|---|---|---|---|---|
| SVR | 50 | 0.1946±0.0711 | 0.2111±0.0777 | 0.2592±0.1108 | 0.4038±0.2174 |
| | 100 | 0.1455±0.0565 | 0.1592±0.0566 | 0.1843±0.0720 | 0.2524± 0.0984 |
| | 300 | 0.0826± 0.0299 | 0.0947±0.0308 | 0.1005±0.0382 | 0.1402±0.0588 |
| EQSVR | 50 | 0.1079±0.0274 | 0.1087±0.0258 | 0.1145±0.0357 | 0.1186±0.0400 |
| | 100 | 0.0981±0.0285 | 0.1007±0.0296 | 0.0965±0.0317 | 0.0986±0.0316 |
| | 300 | 0.0713±0.0254 | 0.0769±0.0237 | 0.0718±0.0237 | 0.0747± 0.0242 |
| PCSVR | 50 | 0.1577±0.0656 | 0.1722±0.0698 | 0.2115±0.0902 | 0.3020±0.1721 |
| | 100 | 0.1151±0.0489 | 0.1294±0.0509 | 0.1513±0.0707 | 0.2122± 0.0936 |
| | 300 | 0.0682±0.0273 | 0.0773±0.0312 | 0.0822±0.0346 | 0.1148±0.0547 |
| PCRSVR | 50 | 0.1007±0.0258 | 0.1009±0.0255 | 0.1112±0.0346 | 0.1145±0.0427 |
| | 100 | 0.0878±0.0275 | 0.0896±0.0301 | 0.0899±0.0322 | 0.0892±0.0326 |
| | 300 | 0.0599±0.0234 | 0.0663±0.0240 | 0.0613±0.0222 | 0.0640±0.0225 |
| PFCSVR | 50 | 0.07698 ±0.0496 | 0.0795±0.0573 | 0.1010±0.0595 | 0.1589±0.1150 |
| | 100 | 0.0645±0.0388 | 0.0616± 0.0372 | 0.0823±0.0482 | 0.1066±0.0764 |
| | 300 | 0.0366±0.0247 | 0.0415±0.0260 | 0.0451±0.0279 | 0.0559±0.0339 |
| PFCRSVR | 50 | 0.0903±0.0311 | 0.0915±0.0294 | 0.1027±0.0374 | 0.1081±0.0461 |
| | 100 | 0.0697±0.0276 | 0.0687±0.0292 | 0.0718±0.0321 | 0.0716±0.0315 |
| | 300 | 0.0395±0.0215 | 0.0405±0.0207 | 0.0381±0.0192 | 0.035±0.0189 |

**Table 3. The summary statistics (mean±S.E) of MSE of regression coefficients regarding proposed and other studied estimators for $p = 5$ and $\rho = 0.9$.**

| Method | Sample size n | 0% | Contamination Levels 5% | 15% | 30% |
|---|---|---|---|---|---|
| SVR | 50 | 0.0977±0.0675 | 0.1189±0.0806 | 0.1731±0.1416 | 0.3775±0.4253 |
| | 100 | 0.0576±0.0410 | 0.0678±0.0417 | 0.0942±0.0668 | 0.171±0.1313 |
| | 300 | 0.0195± 0.0146 | 0.0243± 0.0145 | 0.0289±0.0212 | 0.0578±0.0471 |
| EQSVR | 50 | 0.0134±0.0069 | 0.0138± 0.0061 | 0.0147±0.0079 | 0.0160±0.0108 |
| | 100 | 0.0138±0.0083 | 0.0147±0.0093 | 0.0128±0.0077 | 0.0126±0.0067 |
| | 300 | 0.0104±0.0076 | 0.0117± 0.0070 | 0.0101±0.0063 | 0.0101±0.0059 |
| PCSVR | 50 | 0.0488±0.0507 | 0.0565± 0.0542 | 0.0866±0.0946 | 0.1907±0.2826 |
| | 100 | 0.0352±0.0314 | 0.0421±0.0333 | 0.0609±0.0564 | 0.1064±0.0908 |
| | 300 | 0.0135±0.0104 | 0.0174± 0.0130 | 0.0200±0.0184 | 0.0388±0.0356 |
| PCRSVR | 50 | 0.0114±0.0065 | 0.0118± 0.0058 | 0.0127±0.0076 | 0.0141±0.0103 |
| | 100 | 0.0104±0.0068 | 0.0115± 0.0084 | 0.0107±0.0071 | 0.0101±0.0064 |
| | 300 | 0.0074±0.0057 | 0.0088± 0.0062 | 0.0076±0.0051 | 0.0072± 0.0050 |
| PFCSVR | 50 | 0.0181±0.0291 | 0.0213± 0.0367 | 0.0321±0.0394 | 0.0749± 0.1105 |
| | 100 | 0.0139±0.0193 | 0.0128± 0.0167 | 0.0214±0.0280 | 0.0345± 0.0469 |
| | 300 | 0.0047±0.0063 | 0.0057± 0.0065 | 0.0066±0.0084 | 0.0096± 0.0115 |
| PFCRSVR | 50 | 0.0085±0.0059 | 0.0087± 0.0055 | 0.0108±0.0073 | 0.0127± 0.0104 |
| | 100 | 0.0062±0.0054 | 0.0069± 0.0053 | 0.0066±0.0060 | 0.0065±0.0048 |
| | 300 | 0.0031±0.0036 | 0.0036±0.0036 | 0.0029±0.0029 | 0.0021± 0.0022 |

**Table 4. The summary statistics (mean±S.E) of MAE of regression coefficients regarding proposed and other studied estimators for $p=5$ and $\rho=0.9$.**

| Method | Sample size | Contamination levels | | | |
|---|---|---|---|---|---|
| | $n$ | 0% | 5% | 15% | 30% |
| SVR | 50 | 0.2476± 0.0906 | 0.2736±0.1009 | 0.3264± 0.1384 | 0.4706± 0.1108 |
| | 100 | 0.1920± 0.0735 | 0.2097±0.0749 | 0.2427± 0.0967 | 0.3286± 0.0720 |
| | 300 | 0.1122± 0.0406 | 0.1273±0.0423 | 0.1352± 0.0514 | 0.1885± 0.0382 |
| EQSVR | 50 | 0.0948± 0.0263 | 0.0971± 0.0240 | 0.0996± 0.0336 | 0.1045± 0.0357 |
| | 100 | 0.0935± 0.0299 | 0.0970±0.0317 | 0.0889± 0.0285 | 0.0923± 0.0317 |
| | 300 | 0.0813± 0.0291 | 0.0858± 0.0284 | 0.0805± 0.0281 | 0.0812± 0.0237 |
| PCSVR | 50 | 0.1677± 0.0886 | 0.1815± 0.0888 | 0.2239± 0.1153 | 0.3099± 0.0902 |
| | 100 | 0.1458± 0.0676 | 0.1609± 0.0689 | 0.1846± 0.0935 | 0.2527± 0.0707 |
| | 300 | 0.0916± 0.0380 | 0.1038± 0.0440 | 0.1101± 0.0477 | 0.1521± 0.0346 |
| PCRSVR | 50 | 0.0886± 0.0268 | 0.0903±0.0261 | 0.0941± 0.0332 | 0.0999± 0.0346 |
| | 100 | 0.0810± 0.0283 | 0.0852± 0.0304 | 0.0802± 0.0284 | 0.0807± 0.0322 |
| | 300 | 0.0677± 0.0260 | 0.0732± 0.0290 | 0.0694± 0.0269 | 0.0679± 0.0222 |
| PFCSVR | 50 | 0.0932±0.0671 | 0.0981± 0.0722 | 0.1285± 0.0813 | 0.1824± 0.0595 |
| | 100 | 0.0821±0.0552 | 0.0789± 0.0529 | 0.1032± 0.0684 | 0.1270± 0.0482 |
| | 300 | 0.0473± 0.0339 | 0.0536± 0.0364 | 0.0580± 0.0396 | 0.0708± 0.0279 |
| PFCRSVR | 50 | 0.0796± 0.0311 | 0.0817±0.0286 | 0.0904± 0.0354 | 0.0970± 0.0374 |
| | 100 | 0.0639± 0.0278 | 0.0687± 0.0264 | 0.0642± 0.0295 | 0.0670± 0.0321 |
| | 300 | 0.0409± 0.0242 | 0.0445±0.0239 | 0.0414± 0.0221 | 0.0346± 0.0192 |

**Table 5. The summary statistics (mean±S.E) of MSE of regression coefficients regarding proposed and other studied estimators for $p=5$ and $\rho=0.99$.**

| Method | Sample size | Contamination levels | | | |
|---|---|---|---|---|---|
| | $n$ | 0% | 5% | 15% | 30% |
| SVR | 50 | 0.2202± 0.1591 | 0.2256± 0.1340 | 0.2918± 0.2044 | 0.3433±0.2589 |
| | 100 | 0.2238± 0.1568 | 0.2410± 0.1671 | 0.2954± 0.2159 | 0.3958±0.2518 |
| | 300 | 0.1301± 0.0972 | 0.1572± 0.0913 | 0.1781± 0.1335 | 0.3096±0.2429 |
| EQSVR | 50 | 0.0066± 0.0042 | 0.0066± 0.0039 | 0.0083± 0.0057 | 0.0103±0.0098 |
| | 100 | 0.0056± 0.0031 | 0.0060± 0.0037 | 0.0048± 0.0028 | 0.0053±0.0041 |
| | 300 | 0.0059± 0.0037 | 0.0062± 0.0036 | 0.0056± 0.0035 | 0.0046±0.0026 |
| PCSVR | 50 | 0.0420± 0.0546 | 0.0427± 0.0559 | 0.0824± 0.1071 | 0.1084±0.1403 |
| | 100 | 0.0573± 0.0863 | 0.0525± 0.0681 | 0.0800± 0.1186 | 0.0901±0.1198 |
| | 300 | 0.0304± 0.0434 | 0.0395± 0.0477 | 0.0420± 0.0552 | 0.0501±0.0596 |
| PCRSVR | 50 | 0.0058± 0.0043 | 0.0058± 0.0039 | 0.0076± 0.0057 | 0.0098±0.0100 |
| | 100 | 0.0039± 0.0029 | 0.0044± 0.0033 | 0.0035±0.0026 | 0.0043±0.0040 |
| | 300 | 0.0023± 0.0025 | 0.0023± 0.0022 | 0.0018± 0.0017 | 0.0014± 0.0013 |
| PFCSVR | 50 | 0.0420± 0.0546 | 0.0427± 0.0559 | 0.0824± 0.1071 | 0.1084± 0.1403 |
| | 100 | 0.0573± 0.0863 | 0.0525± 0.0682 | 0.0800± 0.1186 | 0.0901± 0.1198 |
| | 300 | 0.0304± 0.0434 | 0.0395± 0.0477 | 0.0420± 0.0552 | 0.0500± 0.0595 |
| PFCRSVR | 50 | 0.0058± 0.0043 | 0.0058± 0.0039 | 0.0076± 0.0057 | 0.0098± 0.0097 |
| | 100 | 0.0039± 0.0028 | 0.0045± 0.0034 | 0.0035±0.0026 | 0.0043±0.0040 |
| | 300 | 0.0022±0.0024 | 0.0023±0.0022 | 0.0019± 0.0017 | 0.0014± 0.0013 |

**Table 6. The summary statistics (mean±S.E) of MAE of regression coefficients regarding proposed and other studied estimators for $p=5$ and $\rho=0.99$.**

| Method | Sample size | | Contamination levels | | | |
| --- | --- | --- | --- | --- | --- | --- |
| | $n$ | 0% | 5% | 15% | 30% |
| SVR | 50 | 0.3710± 0.1437 | 0.3850± 0.1391 | 0.4303± 0.1708 | 0.4650± 0.1769 |
| | 100 | 0.3811± 0.1430 | 0.3940± 0.1455 | 0.4292± 0.1652 | 0.5115± 0.1755 |
| | 300 | 0.2875± 0.1073 | 0.3255± 0.1080 | 0.3378± 0.1325 | 0.4380± 0.1831 |
| EQSVR | 50 | 0.0710± 0.0273 | 0.0710± 0.0251 | 0.0799± 0.0318 | 0.0885± 0.0416 |
| | 100 | 0.0619± 0.0192 | 0.0636± 0.0213 | 0.0559± 0.0186 | 0.0597± 0.0262 |
| | 300 | 0.0620± 0.0212 | 0.0625± 0.0201 | 0.0600± 0.0197 | 0.0546± 0.0166 |
| PCSVR | 50 | 0.1444± 0.1016 | 0.1425± 0.1032 | 0.2028± 0.1298 | 0.2226± 0.1629 |
| | 100 | 0.1589± 0.1243 | 0.1590± 0.1128 | 0.1922± 0.1505 | 0.2027± 0.1519 |
| | 300 | 0.1166± 0.0916 | 0.1393± 0.0981 | 0.1414± 0.1040 | 0.1586± 0.1072 |
| PCRSVR | 50 | 0.0682± 0.0305 | 0.0694 ±0.0272 | 0.1414± 0.0328 | 0.0881± 0.0427 |
| | 100 | 0.0537± 0.0221 | 0.0577± 0.0242 | 0.0491± 0.0221 | 0.0552± 0.0295 |
| | 300 | 0.0359± 0.0204 | 0.0371± 0.0192 | 0.0330± 0.0169 | 0.0299± 0.0146 |
| PFCSVR | 50 | 0.1444±0.1016 | 0.1425± 0.1032 | 0.2028± 0.1298 | 0.2226± 0.1629 |
| | 100 | 0.1589± 0.1243 | 0.1590± 0.1129 | 0.1922± 0.1505 | 0.2027± 0.1519 |
| | 300 | 0.1166± 0.0916 | 0.1392± 0.0982 | 0.1414± 0.1040 | 0.1586± 0.1072 |
| PFCRSVR | 50 | 0.0682± 0.0305 | 0.0694± 0.0272 | 0.0790± 0.0328 | 0.0881± 0.0425 |
| | 100 | 0.0536± 0.0220 | 0.0577± 0.0243 | 0.0490± 0.0220 | 0.0551± 0.0295 |
| | 300 | 0.0357± 0.0198 | 0.0369± 0.0196 | 0.0338± 0.0172 | 0.0291± 0.0142 |

**Table 7. The summary statistics (mean±S.E) of MSE of regression coefficients regarding proposed and other studied estimators for $p=15$ and $\rho=0.8$.**

| Method | Sample size | | Contamination levels | | | |
| --- | --- | --- | --- | --- | --- | --- |
| | $n$ | 0% | 5% | 15% | 30% |
| SVR | 50 | 0.0835± 0.0397 | 0.1023± 0.0564 | 0.1669± 0.1055 | 0.7963± 0.5228 |
| | 100 | 0.0405± 0.0159 | 0.0446± 0.0172 | 0.0662± 0.0328 | 0.2217± 0.1607 |
| | 300 | 0.0122± 0.0049 | 0.0152± 0.0062 | 0.0207± 0.0076 | 0.0451± 0.0195 |
| EQSVR | 50 | 0.0055± 0.0020 | 0.0056± 0.0018 | 0.0059± 0.0019 | 0.0065± 0.0024 |
| | 100 | 0.0060± 0.0020 | 0.0063± 0.0022 | 0.0061± 0.0019 | 0.0060± 0.0018 |
| | 300 | 0.0059± 0.0021 | 0.0061± 0.0022 | 0.0060± 0.0020 | 0.0063± 0.0025 |
| PCSVR | 50 | 0.0127± 0.0085 | 0.0136± 0.0114 | 0.0210± 0.0186 | 0.0600± 0.0697 |
| | 100 | 0.0067± 0.0044 | 0.0076± 0.0049 | 0.0094± 0.0069 | 0.0227± 0.0217 |
| | 300 | 0.0029± 0.0018 | 0.0033± 0.0021 | 0.0041± 0.0024 | 0.0078± 0.0069 |
| PCRSVR | 50 | 0.0032± 0.0016 | 0.0034± 0.0016 | 0.0038± 0.0018 | 0.0050± 0.0023 |
| | 100 | 0.0023± 0.0013 | 0.0022± 0.0013 | 0.0025±0.0012 | 0.0027± 0.0013 |
| | 300 | 0.0018± 0.0011 | 0.0016± 0.0011 | 0.0018± 0.0011 | 0.0017± 0.0010 |
| PFCSVR | 50 | 0.0027± 0.0030 | 0.0031± 0.0051 | 0.0037± 0.0050 | 0.0097± 0.0163 |
| | 100 | 0.0018± 0.0023 | 0.0017± 0.0022 | 0.0023± 0.0038 | 0.0044± 0.0051 |
| | 300 | 0.0007± 0.0009 | 0.0007± 0.0012 | 0.0009± 0.0010 | 0.0014± 0.0016 |
| PFCRSVR | 50 | 0.0021± 0.0013 | 0.0023± 0.0014 | 0.0026± 0.0016 | 0.0040± 0.0022 |
| | 100 | 0.0010± 0.0007 | 0.0010± 0.0008 | 0.0012±0.0009 | 0.0015± 0.0010 |
| | 300 | 0.0005±0.0005 | 0.0005±0.0006 | 0.0004±0.0004 | 0.0004± 0.0004 |

**Table 8. The summary statistics (mean±S.E) of MAE of regression coefficients regarding proposed and other studied estimators for $p=15$ and $\rho=0.8$.**

| Method | Sample size | | Contamination levels | | |
|---|---|---|---|---|---|
| | $n$ | 0% | 5% | 15% | 30% |
| SVR | 50 | 0.2296± 0.0549 | 0.2495± 0.0611 | 0.3217± 0.1026 | 0.6777± 0.2389 |
| | 100 | 0.1588± 0.0341 | 0.1678± 0.0318 | 0.2030± 0.0522 | 0.3646± 0.1155 |
| | 300 | 0.0877± 0.0180 | 0.0981± 0.0222 | 0.1149± 0.0229 | 0.1690± 0.0376 |
| EQSVR | 50 | 0.0592± 0.0120 | 0.0606± 0.0106 | 0.0622± 0.0103 | 0.0665± 0.0143 |
| | 100 | 0.0614± 0.0115 | 0.0629± 0.0125 | 0.0630± 0.0110 | 0.0620± 0.0100 |
| | 300 | 0.0610± 0.0110 | 0.0619± 0.0130 | 0.0626± 0.0126 | 0.0627± 0.0131 |
| PCSVR | 50 | 0.0867± 0.0303 | 0.0879± 0.0352 | 0.1079± 0.0467 | 0.1783± 0.0968 |
| | 100 | 0.0637± 0.0214 | 0.0683± 0.0238 | 0.0733± 0.0282 | 0.1131± 0.0480 |
| | 300 | 0.0421± 0.0137 | 0.0453± 0.0155 | 0.0503± 0.0149 | 0.0673± 0.0264 |
| PCRSVR | 50 | 0.0462± 0.0121 | 0.0481± 0.0126 | 0.0510± 0.0140 | 0.0606± 0.0165 |
| | 100 | 0.0383± 0.0113 | 0.0369± 0.0113 | 0.0400± 0.0106 | 0.0426± 0.0108 |
| | 300 | 0.0332± 0.0109 | 0.0317± 0.0111 | 0.0333± 0.0109 | 0.0323± 0.0104 |
| PFCSVR | 50 | 0.0374±0.0216 | 0.0380± 0.0281 | 0.0418± 0.0282 | 0.0651± 0.0481 |
| | 100 | 0.0292± 0.0192 | 0.0290± 0.0190 | 0.0329± 0.0228 | 0.0474± 0.0277 |
| | 300 | 0.0183± 0.0119 | 0.0178± 0.0137 | 0.0213± 0.0133 | 0.0268± 0.0164 |
| PFCRSVR | 50 | 0.0401± 0.0139 | 0.0426± 0.0139 | 0.0456± 0.0155 | 0.0576± 0.0186 |
| | 100 | 0.0267± 0.0100 | 0.0270± 0.0102 | 0.0295± 0.0118 | 0.0333± 0.0121 |
| | 300 | 0.0174± 0.0080 | 0.0168± 0.0098 | 0.0160± 0.0082 | 0.0163± 0.0076 |

**Table 9. The summary statistics (mean±S.E) of MSE of regression coefficients regarding proposed and other studied estimators for $p=15$ and $\rho=0.9$.**

| Method | Sample size | | Contamination levels | | |
|---|---|---|---|---|---|
| | $n$ | 0% | 5% | 15% | 30% |
| SVR | 50 | 0.1332± 0.0623 | 0.1534± 0.0725 | 0.2369± 0.1228 | 0.6737± 0.3470 |
| | 100 | 0.0712± 0.0280 | 0.0772± 0.0282 | 0.1127± 0.0534 | 0.3112± 0.1659 |
| | 300 | 0.0227± 0.0092 | 0.0283± 0.0116 | 0.0381± 0.0144 | 0.0812± 0.0352 |
| EQSVR | 50 | 0.0038± 0.0015 | 0.0039± 0.0013 | 0.0040± 0.0014 | 0.0045± 0.0018 |
| | 100 | 0.0048± 0.0017 | 0.0049± 0.0018 | 0.0046± 0.0014 | 0.0043± 0.0014 |
| | 300 | 0.0065± 0.0024 | 0.0066± 0.0024 | 0.0062± 0.0021 | 0.0058± 0.0024 |
| PCSVR | 50 | 0.0212± 0.0135 | 0.0241± 0.0207 | 0.0357± 0.0303 | 0.0892± 0.0952 |
| | 100 | 0.0123± 0.0081 | 0.0137± 0.0091 | 0.0167± 0.0119 | 0.0409± 0.0395 |
| | 300 | 0.0055± 0.0035 | 0.0062± 0.0042 | 0.0075± 0.0043 | 0.0145± 0.0133 |
| PCRSVR | 50 | 0.0024± 0.0013 | 0.0024± 0.0012 | 0.0027± 0.0015 | 0.0035± 0.0017 |
| | 100 | 0.0020± 0.0011 | 0.0019± 0.0011 | 0.0020±0.0011 | 0.0020± 0.0009 |
| | 300 | 0.0020± 0.0013 | 0.0019± 0.0013 | 0.0020± 0.0013 | 0.0018± 0.0011 |
| PFCSVR | 50 | 0.0049± 0.0054 | 0.0057± 0.0102 | 0.0060± 0.0076 | 0.0186± 0.0431 |
| | 100 | 0.0030± 0.0038 | 0.0030± 0.0040 | 0.0040± 0.0068 | 0.0077± 0.0094 |
| | 300 | 0.0013± 0.0017 | 0.0013± 0.0023 | 0.0017± 0.0020 | 0.0025± 0.0029 |
| PFCRSVR | 50 | 0.0015± 0.0010 | 0.0016± 0.0010 | 0.0019± 0.0013 | 0.0029± 0.0017 |
| | 100 | 0.0008± 0.0006 | 0.0008± 0.0007 | 0.0010±0.0009 | 0.0011± 0.0007 |
| | 300 | 0.0005±0.0005 | 0.0005±0.0007 | 0.0005±0.0005 | 0.0004± 0.0004 |

**Table 10. The summary statistics (mean±S.E) of MAE of regression coefficients regarding proposed and other studied estimators for $p=15$ and $\rho=0.9$.**

| Method | Sample size $n$ | 0% | Contamination levels 5% | 15% | 30% |
|---|---|---|---|---|---|
| SVR | 50 | 0.2907± 0.0697 | 0.3074± 0.0666 | 0.3880± 0.1078 | 0.6363± 0.1786 |
| | 100 | 0.2103± 0.0454 | 0.2214± 0.0398 | 0.2645± 0.0667 | 0.4389± 0.1148 |
| | 300 | 0.1198± 0.0251 | 0.1337± 0.0303 | 0.1557± 0.0317 | 0.2271± 0.0502 |
| EQSVR | 50 | 0.0490± 0.0104 | 0.0508± 0.0090 | 0.0514± 0.0101 | 0.0559± 0.0126 |
| | 100 | 0.0552± 0.0105 | 0.0553± 0.0110 | 0.0549± 0.0098 | 0.0523± 0.0089 |
| | 300 | 0.0647± 0.0118 | 0.0643± 0.0133 | 0.0635± 0.0121 | 0.0606± 0.0130 |
| PCSVR | 50 | 0.1121± 0.0380 | 0.1160± 0.0472 | 0.1421± 0.0593 | 0.2194± 0.1092 |
| | 100 | 0.0855± 0.0294 | 0.0914± 0.0317 | 0.0976± 0.0367 | 0.1512± 0.0644 |
| | 300 | 0.0573± 0.0193 | 0.0610± 0.0220 | 0.0674± 0.0205 | 0.0914± 0.0363 |
| PCRSVR | 50 | 0.0393± 0.0115 | 0.0403± 0.0107 | 0.0428± 0.0132 | 0.0516± 0.0145 |
| | 100 | 0.0350± 0.0103 | 0.0339± 0.0100 | 0.0355± 0.0104 | 0.0367± 0.0095 |
| | 300 | 0.0355± 0.0119 | 0.0342± 0.0119 | 0.0353± 0.0111 | 0.0327± 0.0110 |
| PFCSVR | 50 | 0.0488±0.0297 | 0.0488± 0.0404 | 0.0526± 0.0359 | 0.0838± 0.0700 |
| | 100 | 0.0370± 0.0247 | 0.0372± 0.0263 | 0.0418± 0.0314 | 0.0612± 0.0372 |
| | 300 | 0.0240± 0.0162 | 0.0230± 0.0193 | 0.0288± 0.0184 | 0.0344± 0.0224 |
| PFCRSVR | 50 | 0.0331± 0.0130 | 0.0358± 0.0120 | 0.0383 ±0.0140 | 0.0495± 0.0165 |
| | 100 | 0.0238± 0.0094 | 0.0239± 0.0098 | 0.0260± 0.0115 | 0.0282± 0.0107 |
| | 300 | 0.0180± 0.0085 | 0.0174± 0.0101 | 0.0163± 0.0089 | 0.0157± 0.0078 |

**Table 11. The summary statistics (mean±S.E) of MSE of regression coefficients regarding proposed and other studied estimators for $p=15$ and $\rho=0.99$.**

| Method | Sample size $n$ | 0% | Contamination levels 5% | 15% | 30% |
|---|---|---|---|---|---|
| SVR | 50 | 0.2422± 0.0906 | 0.2667± 0.1042 | 0.3302± 0.1125 | 0.4036±0.1575 |
| | 100 | 0.2544± 0.0936 | 0.2742± 0.1038 | 0.3078± 0.1149 | 0.4950±0.1753 |
| | 300 | 0.1528± 0.0636 | 0.1811± 0.0744 | 0.2290± 0.0835 | 0.3870±0.1509 |
| EQSVR | 50 | 0.0012± 0.0008 | 0.0013± 0.0006 | 0.0015± 0.0009 | 0.0023±0.0014 |
| | 100 | 0.0011± 0.0004 | 0.0011± 0.0004 | 0.0011± 0.0004 | 0.0012±0.0005 |
| | 300 | 0.0024± 0.0009 | 0.0021± 0.0008 | 0.0019± 0.0007 | 0.0015±0.0005 |
| PCSVR | 50 | 0.0805± 0.0536 | 0.0784± 0.0516 | 0.0995± 0.0639 | 0.1325±0.0746 |
| | 100 | 0.0634± 0.0436 | 0.0671± 0.0402 | 0.0741± 0.0500 | 0.1244±0.0931 |
| | 300 | 0.0410± 0.0276 | 0.0440± 0.0318 | 0.0520± 0.0299 | 0.0865±0.0737 |
| PCRSVR | 50 | 0.0011± 0.0007 | 0.0011± 0.0006 | 0.0013± 0.0010 | 0.0022±0.0013 |
| | 100 | 0.0006± 0.0004 | 0.0006± 0.0003 | 0.0007±0.0004 | 0.0008±0.0005 |
| | 300 | 0.0009± 0.0006 | 0.0008± 0.0005 | 0.0008± 0.0005 | 0.0006± 0.0003 |
| PFCSVR | 50 | 0.0214± 0.0271 | 0.0192± 0.0275 | 0.0236± 0.0307 | 0.0340± 0.0419 |
| | 100 | 0.0156± 0.0211 | 0.0163± 0.0249 | 0.0203± 0.0280 | 0.0319± 0.0423 |
| | 300 | 0.0095± 0.0125 | 0.0099± 0.0177 | 0.0123± 0.0149 | 0.0169± 0.0215 |
| PFCRSVR | 50 | 0.0009± 0.0007 | 0.0010± 0.0006 | 0.0012± 0.0009 | 0.0021± 0.0013 |
| | 100 | 0.0004± 0.0003 | 0.0004± 0.0003 | 0.0005±0.0004 | 0.0007±0.0005 |
| | 300 | 0.0003±0.0003 | 0.0002±0.0002 | 0.0002±0.0002 | 0.0002± 0.0001 |

**Table 12. The summary statistics (mean±S.E) of MAE of regression coefficients regarding proposed and other studied estimators for $p = 15$ and $\rho = 0.99$.**

| Method | Sample size n | 0% | Contamination levels 5% | 15% | 30% |
|---|---|---|---|---|---|
| SVR | 50 | 0.3955± 0.0784 | 0.4100± 0.0849 | 0.4631± 0.0834 | 0.5067± 0.1038 |
| | 100 | 0.4042± 0.0782 | 0.4187± 0.0769 | 0.4475± 0.0917 | 0.5633± 0.1059 |
| | 300 | 0.3093± 0.0640 | 0.3394± 0.0766 | 0.3821± 0.0762 | 0.4978± 0.0983 |
| EQSVR | 50 | 0.0298± 0.0101 | 0.0307± 0.0095 | 0.0334± 0.0122 | 0.0436± 0.0153 |
| | 100 | 0.0271± 0.0056 | 0.0264± 0.0054 | 0.0268± 0.0067 | 0.0282± 0.0079 |
| | 300 | 0.0390± 0.0074 | 0.0369± 0.0080 | 0.0358± 0.0070 | 0.0309± 0.0056 |
| PCSVR | 50 | 0.2168± 0.0783 | 0.2140± 0.0751 | 0.2429± 0.0863 | 0.2823± 0.0815 |
| | 100 | 0.1941± 0.0686 | 0.2044± 0.0618 | 0.2064± 0.0771 | 0.2692± 0.0936 |
| | 300 | 0.1558± 0.0542 | 0.1604± 0.0612 | 0.1763± 0.0545 | 0.2240± 0.0879 |
| PCRSVR | 50 | 0.0285± 0.0111 | 0.0292± 0.0106 | 0.0325± 0.0134 | 0.0434± 0.0155 |
| | 100 | 0.0210± 0.0070 | 0.0208± 0.0063 | 0.0221± 0.0082 | 0.0251± 0.0094 |
| | 300 | 0.0244± 0.0078 | 0.0218± 0.0074 | 0.0220± 0.0070 | 0.0190± 0.0062 |
| PFCSVR | 50 | 0.0980±0.0682 | 0.0890± 0.0724 | 0.1013± 0.0729 | 0.1234± 0.0826 |
| | 100 | 0.0819± 0.0601 | 0.0813± 0.0659 | 0.0926± 0.0695 | 0.1226± 0.0790 |
| | 300 | 0.0638± 0.0446 | 0.0603± 0.0560 | 0.0738± 0.0523 | 0.0876± 0.0607 |
| PFCRSVR | 50 | 0.0278± 0.0118 | 0.0286± 0.0112 | 0.0321±0.0138 | 0.0434± 0.0159 |
| | 100 | 0.0179± 0.0080 | 0.0179± 0.0071 | 0.0196±0.0095 | 0.0237± 0.0104 |
| | 300 | 0.0142± 0.0062 | 0.0127± 0.006 | 0.0123± 0.0057 | 0.0113± 0.0052 |

for $\rho$ = 0.99. For example, the percentages reduction in MSE of PFCRSVR against PFCSVR are 90%, 99% and 98% for $n$ = 100, 300 and 500, respectively when $p$ = 25 and percent contamination is 30 (see, Fig 4d). Similarly, with the same level of collinearity and contamination, the resulting pattern remains consistent when $p$ = 15, $n$ = 50, 100 and, 300. Consequently,

the maximum reduction in MSE is exhibited up to 99% for proposed techniques over their competitors even with a high concentration of outliers and collinearity. Further, the proposed techniques are also compared with baseline EQSVR and come out to be competent. Because they exhibit a maximum reduction in MSE over all considered simulation settings (see, Figs 2–7 and S1-S3 Figs). Therefore, the results indicate that proposed approaches outperform other competing techniques by overcoming the effects of anomalies and multicollinearity simultaneously.

## 5. Discussion

The results of this study demonstrate the robustness and effectiveness of the proposed regression frameworks (e.g., PCRSVR and PFCRSVR). Extensive simulations reveal that these methods consistently outperform their baseline counterparts (e.g., PCSVR, PFCSVR, and EQSVR). The proposed frameworks excel in handling challenges such as high multicollinearity, anomaly severity, and varying sample and predictor sizes. Both PCRSVR and PFCRSVR achieve significantly lower MSE and MAE values. These results showcase their ability to mitigate the adverse effects of extreme data complexities as well as ill-conditioned predictors. Moreover, the validation using real-life datasets highlights the practical relevance of these approaches. The proposed techniques consistently outperform baseline methods for real-life datasets characterized by high multicollinearity and the presence of outliers. These findings underline the generalizability and effectiveness of PCRSVR and PFCRSVR in tackling real-world regression challenges.

Despite these promising results, certain limitations seek attention. The study focuses solely on normal response variable and vertical outliers. This excludes scenarios involving non-normal responses, such as binary or count data, as well

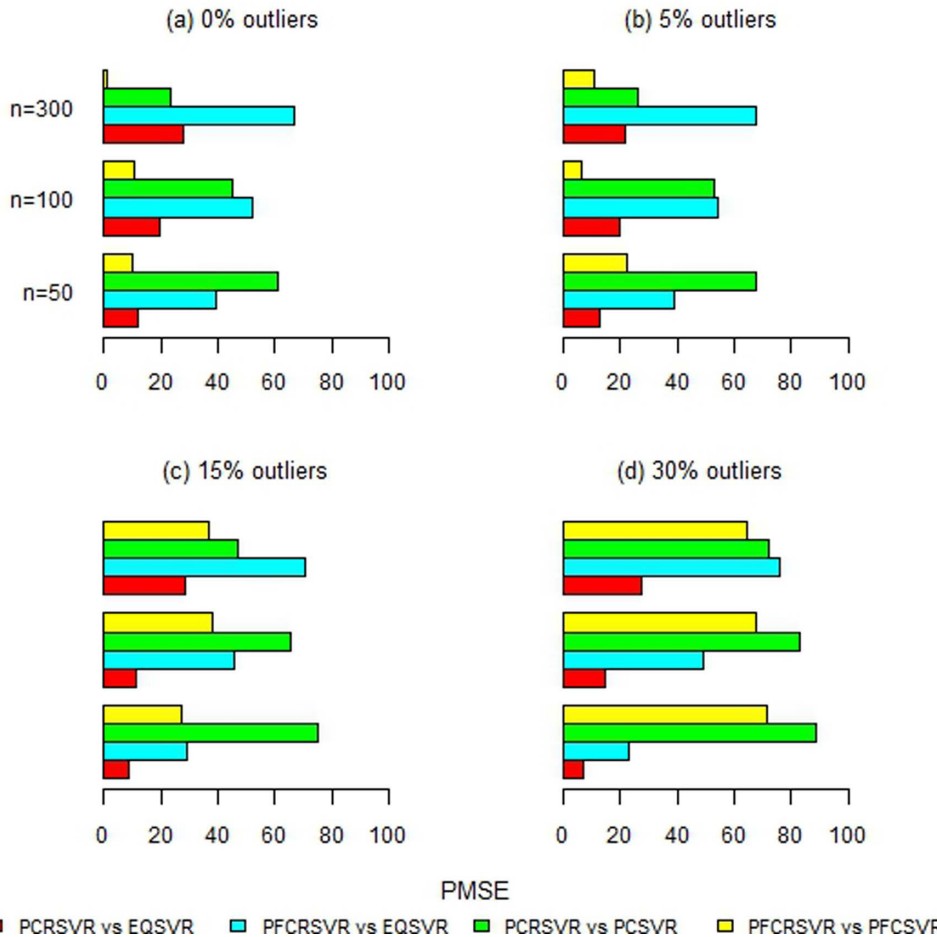

**Fig 2. Improved percentage reduction in MSE of proposed techniques against their competitors with different levels of contamination for p=5 and ρ=0.8.**

as leverage points in the predictor space. These restrictions may limit the applicability of the methods in domains with more diverse data characteristics. Future research could focus on extending the frameworks to handle response variables from the exponential family. Modifications could also address leverage points to improve the methods' robustness. Expanding the scope in these directions would enhance the utility and adaptability of the proposed techniques.

Another limitation stems from the nature of the datasets analyzed. The study predominantly focuses on cases where the number of observations exceed the number of predictors. While this condition is common in many regression applications, it does not account for high-dimensional settings where predictors outnumber observations. In such scenarios, standard dimensionality reduction techniques, like principal components, may not perform optimally. Future work could adapt the frameworks to high-dimensional datasets. This could involve advanced strategies such as sparsity-inducing penalties or tailored regularization techniques. Addressing these gaps would extend the applicability of these methods to fields like genomics, text mining, and image analysis.

## 6. Real-life data application

This section demonstrates the performance of the proposed techniques using the pollution and Longley datasets. These datasets have been widely analyzed in previous research (e.g., [39,47–49]). According to the literature, these datasets are

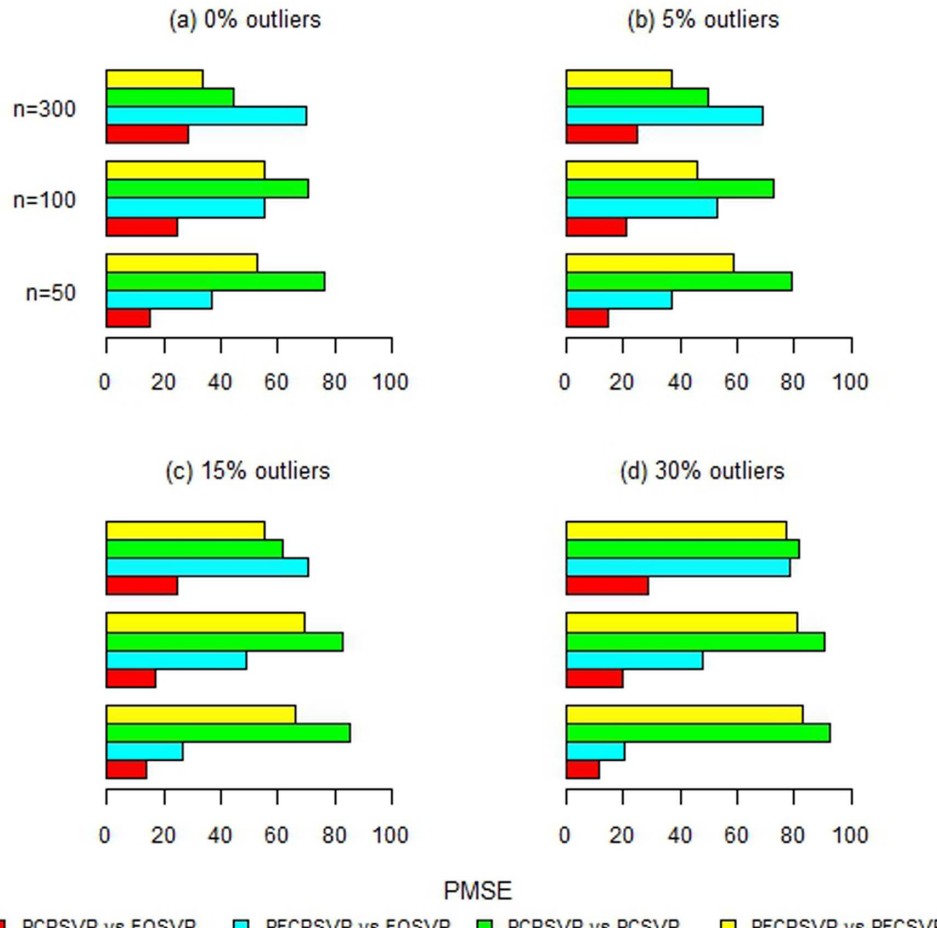

**Fig 3. Improved percentage reduction in MSE of proposed techniques against their competitors with different levels of contamination for p = 5 and ρ = 0.9.**

known for having ill-conditioned predictors and extreme observations. Therefore, these real-life datasets are suitable for evaluating methods that address these issues efficiently.

In the pollution dataset, the outcome variable is the age-adjusted mortality rate per 100,000, which depends on fifteen explanatory variables. A detailed description of these covariates is available in prior studies (e.g., [47,49]). Application of the least square method reveals a high degree of multicollinearity, with variance inflation factors of 98.6 for $x_{12}$ and 104.9 for $x_{13}$. The strength of correlation among predictors is illustrated in Fig 8a. Residual analysis is also conducted to identify extreme observations. Normal QQ plots of residuals and Cook's distances indicate that observations 2, 29, 32, 37, 48, 57, and 59 are outliers (see Fig 9a and 9b). These findings confirm that the dataset exhibits both multicollinearity and outliers, making it an appropriate example for testing the proposed techniques.

The regression coefficients of the predictors are estimated using all examined modeling frameworks, with results presented in Table 13. For performance evaluation, the standard errors of the bootstrap regression estimates are calculated for each technique, and the mean standard errors are reported (see, Table 13). The results indicate that the proposed techniques yield lower Mean Standard Errors of Bootstrap Estimates (MSEBE) compared to competing methods. Notably, the PFCRSVR method outperforms all other approaches, achieving the lowest mean standard error of the bootstrap regression estimates.

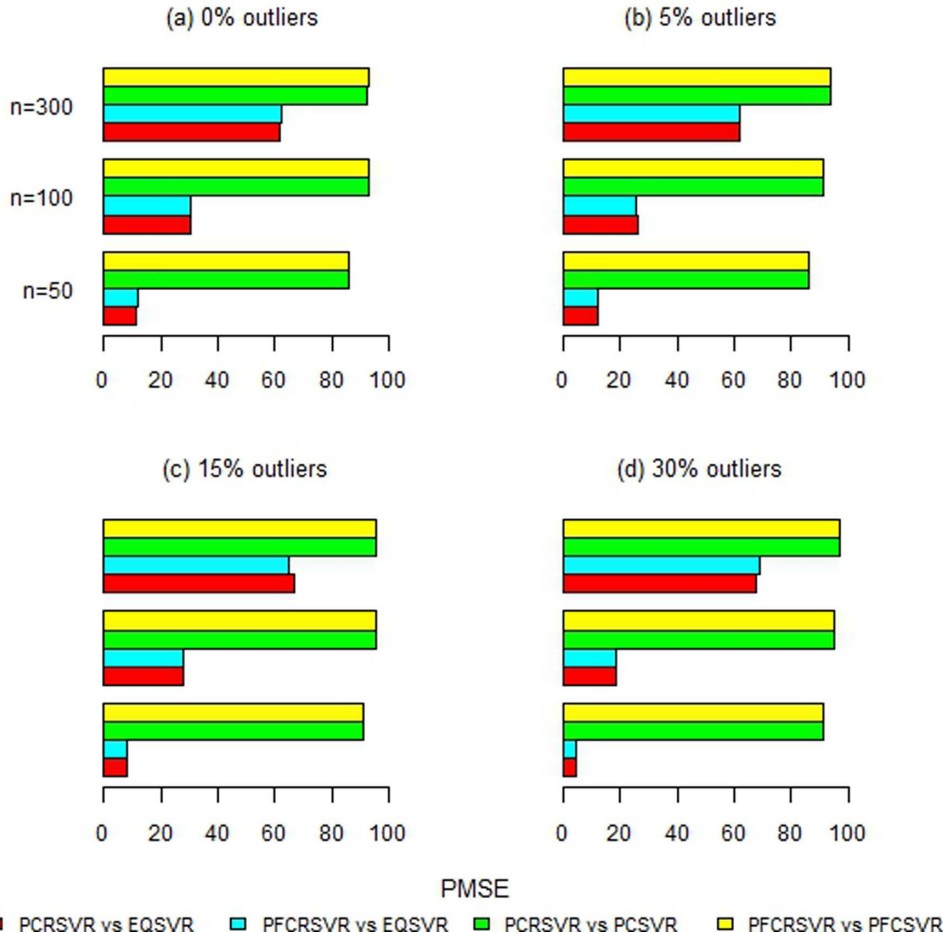

**Fig 4. Improved percentage reduction in MSE of proposed techniques against their competitors with different levels of contamination for p = 5 and ρ = 0.99.**

Further, the Longley dataset [50] comprise five predictors with the objective of modeling total derived employment ($y$). These predictors include the Gross National Product (GNP) implicit price deflator ($x_1$), GNP ($x_2$), unemployment rate ($x_3$), size of the armed forces ($x_4$), and non-institutional population aged fourteen years and older ($x_5$). Prior research has highlighted the significant effect of multicollinearity and the presence of outliers within this dataset [40].

This is indicated by a high condition index of 43,275 and variance inflation factors of 5,209.50, 306.50, 2,825.30, 37.74, and 39.90 for the predictors. Fig 8b provides an illustration of the correlations among the predictors, highlighting the extent of multicollinearity. Additionally, Fig 10 includes a QQ-plot of residuals and Cook's distances, which reveal data points 6, 10, 12, 14, and 16 as notable outliers.

Regression parameters are estimated for both the proposed method and existing approaches and their results are presented in Table 14. To further assess estimation accuracy, bootstrap coefficients are estimated. The $\text{MSEBE}(\hat{\beta})$ for each method is also reported in Table 14. The proposed methods demonstrate favourable performance compared to competing techniques, achieving the lowest $\text{MSEBE}(\hat{\beta})$ value.

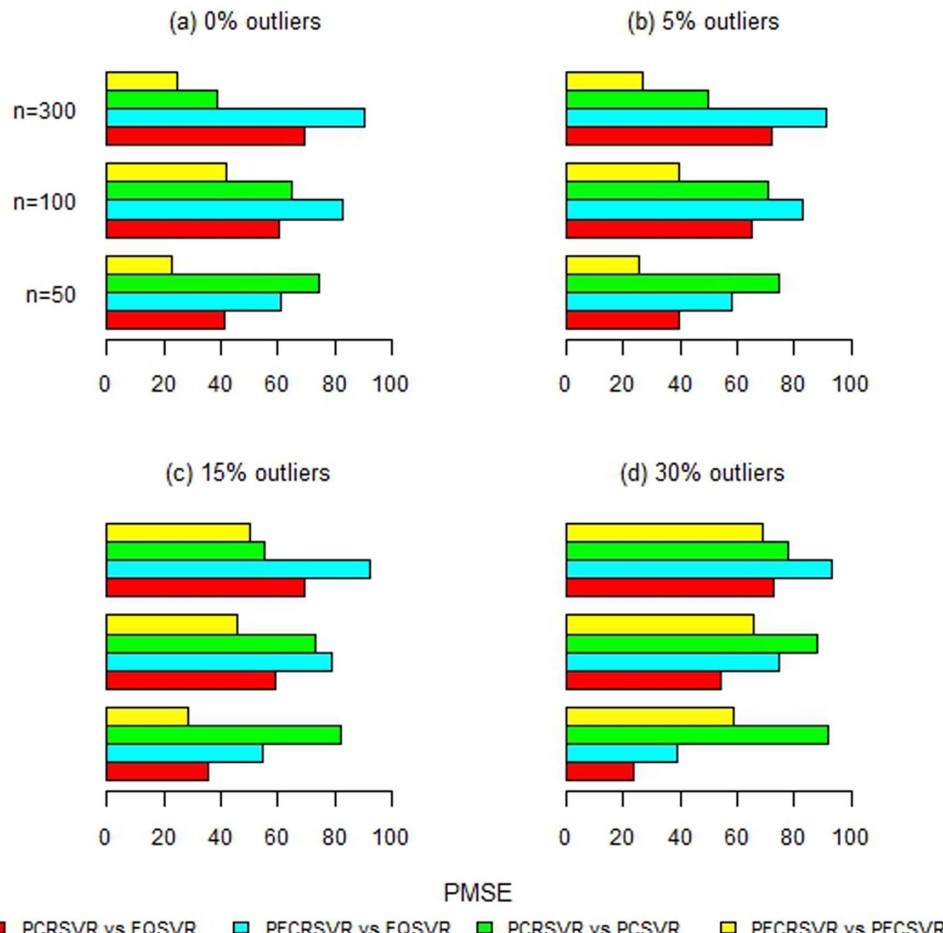

**Fig 5. Improved percentage reduction in MSE of proposed techniques against their competitors with different levels of contamination for p = 15 and ρ = 0.8.**

## 7. Conclusion

This research advances a robust regression framework by addressing core challenges, including multicollinearity, outliers, and high-dimensional data, which constrains the effectiveness of classical SVR. By proposing frameworks that incorporate PCs and PFCs, the study not only tackles critical issues but also broadens the scope of SVR's applicability to more intricate and irregular data environments.

Moreover, the findings highlight a broader paradigm shift in developing robust regression approaches to tackle real-world challenges. Fields such as finance, healthcare, and environmental science frequently face complex data structures. These complexities often compromise the accuracy of predictive models. The proposed innovations provide optimal benefits in these fields. The ability to address ill-conditioned predictors and neutralize the effects of anomalies positions these frameworks as transformative tools for practitioners.

## Supporting information

**S1 File.** S1 Table. The summary statistics (mean±S.E) of MSE of regression coefficients regarding proposed and other studied estimators for p = 25 and ρ = 0.9. S2 Table. The summary statistics (mean±S.E) of MAE of regression coefficients regarding proposed and other studied estimators for $p = 25$ and $\rho = 0.9$. S3 Table. The summary statistics (mean±S.E)

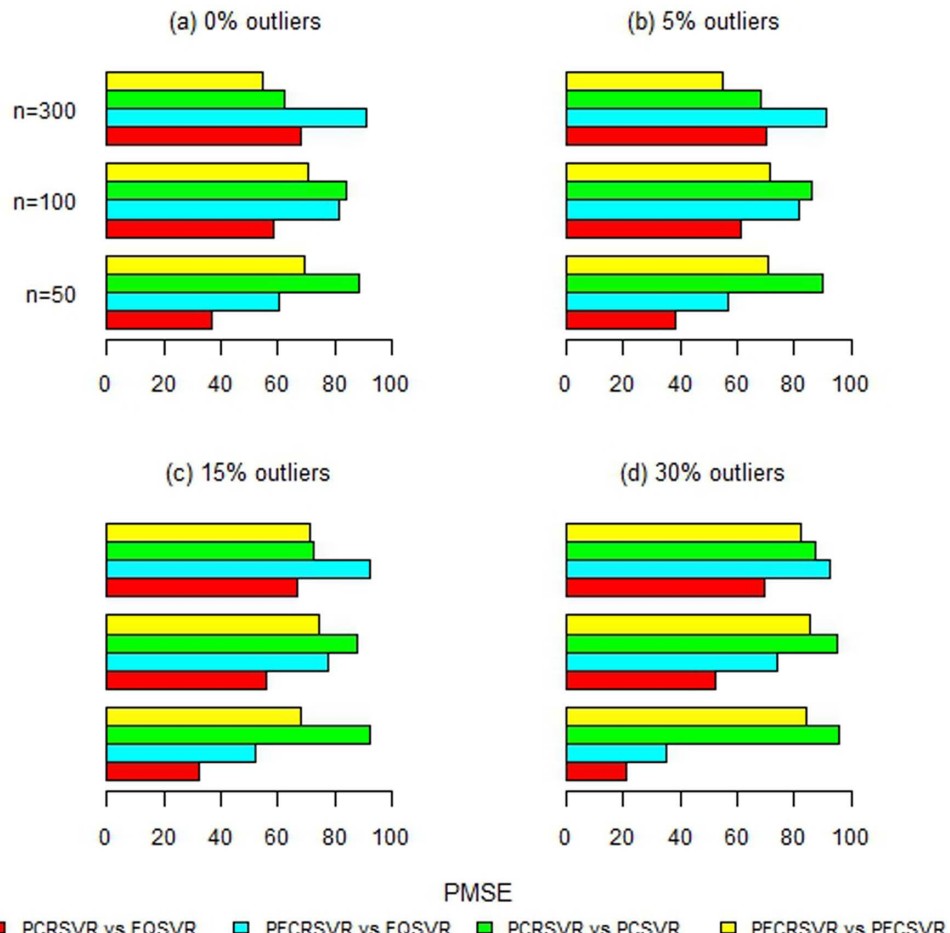

Fig 6. **Improved percentage reduction in MSE of proposed techniques against their competitors with different levels of contamination for p = 15 and ρ = 0.9.**

of MSE of regression coefficients regarding proposed and other studied estimators for $p = 25$ and $\rho = 0.8$. S4 Table. The summary statistics (mean ± S.E) of MAE of regression coefficients regarding proposed and other studied estimators for $p = 25$ and $\rho = 0.8$. S5 Table. The summary statistics (mean ± S.E) of MSE of regression coefficients regarding proposed and other studied estimators for $p = 25$ and $\rho = 0.99$. S6 Table. The summary statistics (mean ± S.E) of MAE of regression coefficients regarding proposed and other studied estimators for $p = 25$ and $\rho = 0.99$. S7 Table. A list of abbreviations used in the paper.
(PDF)

**S1 Fig. Improved percentage reduction in MSE of proposed techniques against their competitors with different levels of contamination for $p = 25$ and $\rho = 0.9$.**
(TIFF)

**S2 Fig. Improved percentage reduction in MSE of proposed techniques against their competitors with different levels of contamination for $p = 25$ and $\rho = 0.9$.**
(TIFF)

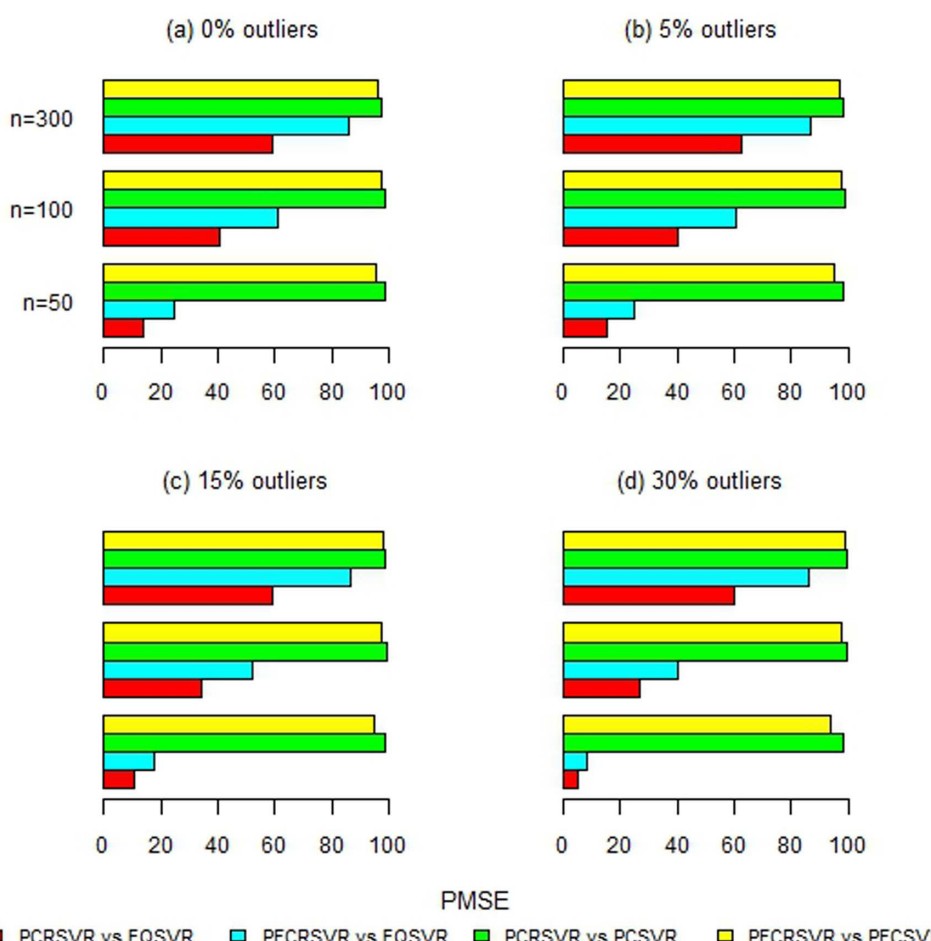

**Fig 7. Improved percentage reduction in MSE of proposed techniques against their competitors with different levels of contamination for p = 15 and ρ = 0.99.**

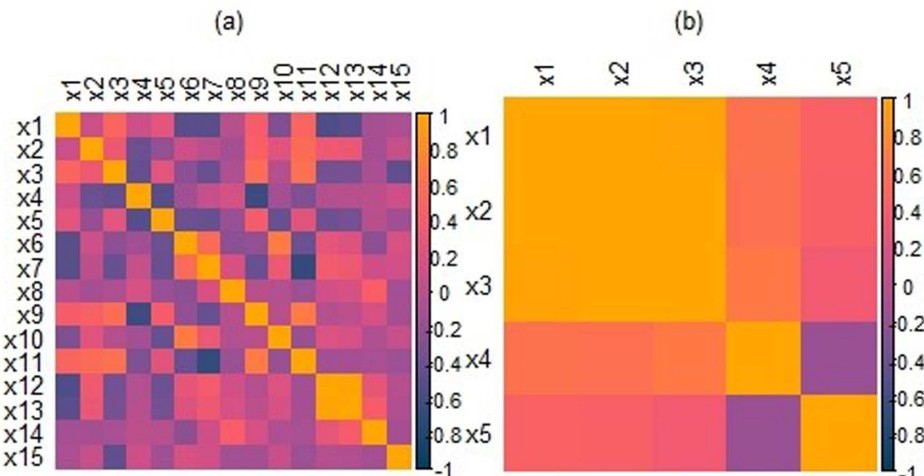

**Fig 8. The graphical presentation of correlations among predictors of the pollution data (a) and the Longley data (b).**

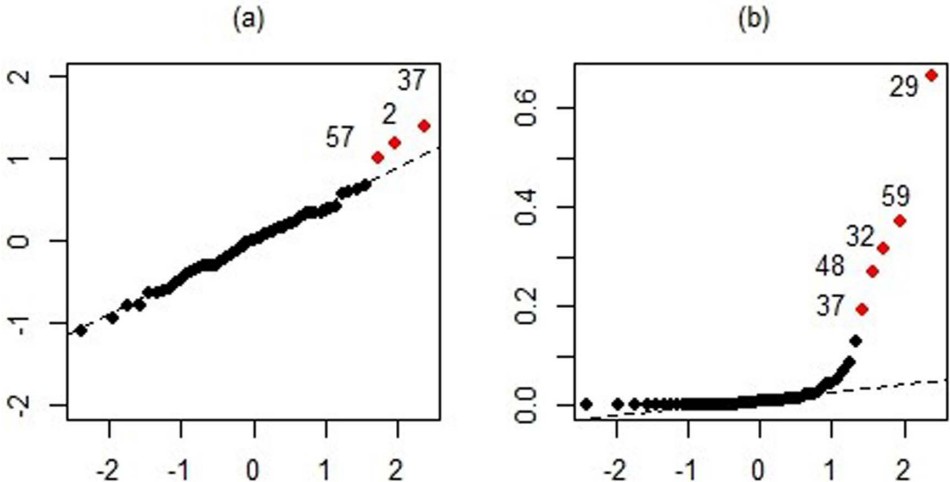

**Fig 9. The normal QQ-plot of residuals (a) and Cook's distances (b) of pollution data.**

**Table 13. The estimated regression coefficients and MSEBE for proposed and competing techniques using the Pollution dataset.**

| Method | SVR | EQSVR | PCSVR | PCRSVR | PFCSVR | PFCRSVR |
|---|---|---|---|---|---|---|
| $\beta_1$ | 12.8303 | −0.00287 | 3.2759 | −0.01121 | 7.00588 | 0.0064438 |
| $\beta_2$ | −2.61895 | −0.01070 | 1.5316 | 0.001958 | −0.7923 | 0.003978 |
| $\beta_3$ | 0.43041 | −0.00551 | 3.249086 | −0.00717 | 3.6237 | 0.005631 |
| $\beta_4$ | −3.8600 | −0.00801 | −1.3484 | −0.00424 | −2.1082 | 0.00556 |
| $\beta_5$ | 3.4752 | 0.01450 | 3.3785 | −0.00189 | 4.7249 | 0.006681 |
| $\beta_6$ | −5.24980 | −0.00341 | −8.5930 | 0.00746 | −6.9202 | 0.00768 |
| $\beta_7$ | −5.3095 | −0.01066 | −4.2662 | 0.00814 | −5.8695 | 0.005391 |
| $\beta_8$ | 10.51762 | −0.000673 | 5.7833 | −0.00052 | 3.6557 | 0.0033005 |
| $\beta_9$ | 14.1016 | 0.00223 | 6.0058 | −0.00132 | 8.52843 | 0.011856 |
| $\beta_{10}$ | −1.61165 | −0.00377 | −5.9476 | 0.00420 | −4.0628 | 0.001917 |
| $\beta_{11}$ | 1.3120 | −0.00767 | 5.3982 | −0.00578 | 5.3878 | 0.008137 |
| $\beta_{12}$ | −3.6570 | −0.00650 | 2.7678 | 0.01094 | −2.5277 | −0.001200 |
| $\beta_{13}$ | −0.0682 | −0.00540 | 4.0133 | 0.01062 | −1.15226 | 0.000035 |
| $\beta_{14}$ | 14.1169 | 0.00827 | 7.7954 | 0.00391 | 5.79671 | 0.006070 |
| $\beta_{15}$ | 1.8642 | 0.00177 | −4.4385 | 0.00238 | −1.28665 | −3.1749 |
| **MSEBE($\hat{\beta}$)** | 2.6299 | 0.05213 | 2.7732 | 0.04699 | 1.8994 | 0.03119 |

**S3 Fig. Improved percentage reduction in MSE of proposed techniques against their competitors with different levels of contamination for $p = 25$ and $\rho = 0.99$.**
(TIFF)

## Author contributions

**Conceptualization:** Maryam Ilyas.

**Formal analysis:** Aiman Tahir.

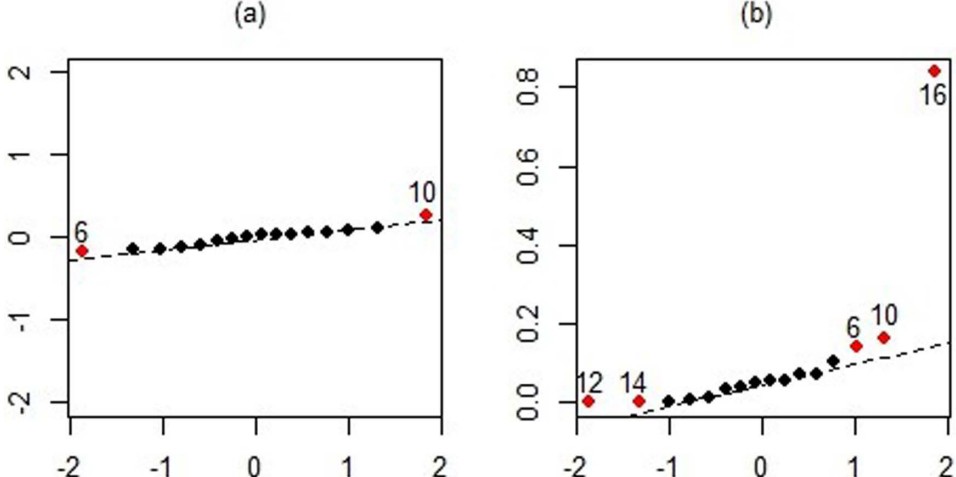

**Fig 10. The normal QQ-plot of residuals (a) and Cook's distances (b) of Longley data.**

**Table 14. The estimated regression coefficients and MSEBE for proposed and competing techniques using the Longley dataset.**

| Method | SVR | EQSVR | PCSVR | PCRSVR | PFCSVR | PFCRSVR |
|---|---|---|---|---|---|---|
| $\beta_1$ | 12.8452 | 0.01316 | 10.5864 | 0.00229 | 13.3980 | 0.0000477 |
| $\beta_2$ | −3.5631 | −0.00220 | −1.2951 | 0.00936 | −1.5407 | 0.0085144 |
| $\beta_3$ | 5.9404 | 0.00540 | 6.9416 | −0.00251 | 6.9176 | 0.0041304 |
| $\beta_4$ | −6.0032 | 0.00540 | −4.0788 | 0.00676 | −4.0121 | −0.006546 |
| $\beta_5$ | 9.0199 | 0.00183 | 11.1796 | −0.01182 | 9.0233 | 0.0041898 |
| **MSEBE** ($\hat{\beta}$) | 3.1002 | 0.0240 | 3.0957 | 0.0237 | 3.0623 | 0.0229 |

**Methodology:** Aiman Tahir, Maryam Ilyas.

**Validation:** Maryam Ilyas.

**Writing – original draft:** Aiman Tahir.

**Writing – review & editing:** Maryam Ilyas.

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
