## [Decision Letter · Decision Letter 0]

21 Jan 2025

PONE-D-24-60709Principal fitted component framework for robust support vector regression based on bounded loss: A simulation study with potential applicationsPLOS ONE

Dear Dr. Tahir,

Thank you for submitting your manuscript to PLOS ONE. After careful consideration, we feel that it has merit but does not fully meet PLOS ONE’s publication criteria as it currently stands. Therefore, we invite you to submit a revised version of the manuscript that addresses the points raised during the review process.

We look forward to receiving your revised manuscript.

Kind regards,

Mohamed R. Abonazel, Ph.D.

Academic Editor

PLOS ONE

Journal Requirements:

Reviewers' comments:

Reviewer's Responses to Questions

**Comments to the Author**

1. Is the manuscript technically sound, and do the data support the conclusions?

Reviewer #1: Yes

2. Has the statistical analysis been performed appropriately and rigorously? 

Reviewer #1: Yes

3. Have the authors made all data underlying the findings in their manuscript fully available?

Reviewer #1: No

4. Is the manuscript presented in an intelligible fashion and written in standard English?

Reviewer #1: Yes

5. Review Comments to the Author

Reviewer #1: TITLE: Principal fitted component framework for robust support vector regression based on bounded loss: A simulation study with potential applications

Manuscript Number: PONE-D-24-60709_reviewer

Type of manuscript: Original Article

Summary

In this paper, the authors present two modelling frameworks: Principal Component Robust Support Vector Regression (PCRSVR) and Principal Fitted Component Robust Support Vector Regression (PFCRSVR). These frameworks are expected to enhance estimation accuracy by integrating Exponential Quantile SVR with principal components and fitted components, effectively managing anomalies, ill-conditioned predictors, and high-dimensional data, as supported by simulation studies and real-life applications.

The novelty of the approach lies in how these elements are integrated to enhance robustness against anomalies and ill-conditioned predictors while maintaining effective performance in high-dimensional data settings. The specific focus on a “bounded loss” framework further indicates a unique contribution, as this may address issues of sensitivity in traditional SVR methods that can be adversely affected by outliers or noise. However, the manuscript requires some improvements. Some general comments are:

Comments

1. Arrange the keywords in alphabetic order.

2. Provide a list of abbreviations used in the paper.

3. At the end of all figure and table captions, insert a full stop.

4. List the study’s highlights at the end of the “Introduction” section.

5. Software packages used should be cited to acknowledge the author(s).

6. I suggest you create a separate “Discussion” section. In the discussion section, the authors should present their findings and their main implications, highlight their study’s limitations, and briefly mention some precise directions they intend to follow in their future research work. Can the authors mention how much of their research is influenced by the data they used or to which extent the methodology used within the developed research can be easily applied to other situations when the datasets differ? In this way, the authors could highlight the generalisation capability of their approach to justify a wider contribution to the current state of the art.

7. In the “Conclusion” section, authors should avoid summarising the aspects they have already stated in the body of the manuscript. Instead, they should interpret their findings at a higher level of abstraction than in the previous sections of the manuscript. The authors should highlight whether or to what extent they have addressed the necessity identified within the “Introduction” section (the identified gap). The authors should avoid restating everything they did once again. However, instead, they should emphasise what their findings mean to the readers, making the “Conclusions” section interesting and memorable to them. The authors should not restate what they have done or what the article does. They should focus instead on what they have discovered and, most importantly, on what their findings mean.

8. The authors point out that a limitation of the study is the use of a narrow dataset range, which affects the generalizability of the findings. This could be expanded further to discuss how the specific characteristics of these datasets might skew results or what types of datasets could serve as valid tests for these frameworks.

9. The authors mention that the methods were tested only on continuous response variables assuming normality. This crucial point could have been elaborated on by discussing the implications of such limitations, such as the potential mismatch between method appropriateness and real-world data distribution.

10. While the focus on outliers in the response variable is commendable, the review could benefit from a deeper exploration of how leverage points in predictor space might impact the performance of these methods. This omission points to a significant blind spot in assessing the overall robustness of the proposed frameworks.

11. The assertion that the techniques are limited to scenarios where the number of observations exceeds the number of predictors is an important limitation that could be explored further. A brief discussion on the implications of applying these methods in high-dimensional settings could clarify the next research steps.

12. In the spirit of reproducible research, the authors should provide both the data and the codes. These can be deposited on GitHub, for example.

6. PLOS authors have the option to publish the peer review history of their article (what does this mean? ). If published, this will include your full peer review and any attached files.

**Do you want your identity to be public for this peer review?** For information about this choice, including consent withdrawal, please see our Privacy Policy .

Reviewer #1: No

---

## [Author Response · Author response to Decision Letter 1]

30 Jan 2025

Response to Reviews

Academic editor:

1. Comment: Please ensure that your manuscript meets PLOS ONE's style requirements, including those for file naming.

Response: Revised manuscript is prepared according to PLOS ONE's style requirements.

2. Comment: Please note that PLOS ONE has specific guidelines on code sharing for submissions in which author-generated code underpins the findings in the manuscript. In these cases, we expect all author-generated code to be made available without restrictions upon publication of the work. Please review our guidelines at https://journals.plos.org/plosone/s/materials-and-software-sharing#loc-sharing-code and ensure that your code is shared in a way that follows best practice and facilitates reproducibility and reuse.

Response: The relevant codes are deposited on Github and the link is added in manuscript.

3. Comment: PLOS requires an ORCID iD for the corresponding author in Editorial Manager on papers submitted after December 6th, 2016. Please ensure that you have an ORCID iD and that it is validated in Editorial Manager. To do this, go to ‘Update my Information’ (in the upper left-hand corner of the main menu), and click on the Fetch/Validate link next to the ORCID field. This will take you to the ORCID site and allow you to create a new iD or authenticate a pre-existing iD in Editorial Manager.

Response: The ORCID iD for the corresponding author is validated in Editorial Manager.

4. Comment: PLOS authors have the option to publish the peer review history of their article. If published, this will include your full peer review and any attached files.

Response: No, the peer review history is not required.

Reviewer - 1:

Comments to the Author

TITLE: Principal fitted component framework for robust support vector regression based on bounded loss: A simulation study with potential applications

Manuscript Number: PONE-D-24-60709_reviewer

Type of manuscript: Original Article

Summary

In this paper, the authors present two modelling frameworks: Principal Component Robust Support Vector Regression (PCRSVR) and Principal Fitted Component Robust Support Vector Regression (PFCRSVR). These frameworks are expected to enhance estimation accuracy by integrating Exponential Quantile SVR with principal components and fitted components, effectively managing anomalies, ill-conditioned predictors, and high-dimensional data, as supported by simulation studies and real-life applications.

The novelty of the approach lies in how these elements are integrated to enhance robustness against anomalies and ill-conditioned predictors while maintaining effective performance in high-dimensional data settings. The specific focus on a “bounded loss” framework further indicates a unique contribution, as this may address issues of sensitivity in traditional SVR methods that can be adversely affected by outliers or noise. However, the manuscript requires some improvements. Some general comments are:

Response: We are grateful for the encouraging comments.

Some comments:

1. Comment: Arrange the keywords in alphabetic order.

Response: The keywords are arranged in alphabetic order. The changes are highlighted in a separate file.

2. Comment: Provide a list of abbreviations used in the paper.

Response: A list of abbreviations used in this paper has been added in supporting information.

3. Comment: At the end of all figure and table captions, insert a full stop.

Response: A full stop is inserted at the end of all figure and table captions.

4. Comment: List the study’s highlights at the end of the “Introduction” section.

Response: The text is added considering above suggestions. The changes are highlighted in the revised file.

5. Comment: Software packages used should be cited to acknowledge the author(s).

Response: Software packages are cited considering above suggestions. The changes are highlighted in the revised file.

6- Comment: I suggest you create a separate “Discussion” section. In the discussion section, the authors should present their findings and their main implications, highlight their study’s limitations, and briefly mention some precise directions they intend to follow in their future research work. Can the authors mention how much of their research is influenced by the data they used or to which extent the methodology used within the developed research can be easily applied to other situations when the datasets differ? In this way, the authors could highlight the generalisation capability of their approach to justify a wider contribution to the current state of the art.

Response: We thank the reviewer for the valuable suggestion. A separate "Discussion" section has been created to better articulate the findings, their implications, the study's limitations, and directions for future research.

7- Comment: In the “Conclusion” section, authors should avoid summarizing the aspects they have already stated in the body of the manuscript. Instead, they should interpret their findings at a higher level of abstraction than in the previous sections of the manuscript. The authors should highlight whether or to what extent they have addressed the necessity identified within the “Introduction” section (the identified gap). The authors should avoid restating everything they did once again. However, instead, they should emphasise what their findings mean to the readers, making the “Conclusions” section interesting and memorable to them. The authors should not restate what they have done or what the article does. They should focus instead on what they have discovered and, most importantly, on what their findings mean.

Response: The “Conclusion” section is updated according to the above suggestions.

8- Comment: The authors point out that a limitation of the study is the use of a narrow dataset range, which affects the generalizability of the findings. This could be expanded further to discuss how the specific characteristics of these datasets might skew results or what types of datasets could serve as valid tests for these frameworks.

Response: In the “discussion” section we address how the specific characteristics of the datasets used might influence the results. Additionally, we provide suggestions on the types of datasets that could serve as valid tests for further evaluation of the proposed frameworks.

9- Comment: The authors mention that the methods were tested only on continuous response variables assuming normality. This crucial point could have been elaborated on by discussing the implications of such limitations, such as the potential mismatch between method appropriateness and real-world data distribution.

Response: We have expanded the discussion to address how this limitation could impact the applicability of the proposed methods and suggested directions for future work to address this issue.

10- Comment: While the focus on outliers in the response variable is commendable, the review could benefit from a deeper exploration of how leverage points in predictor space might impact the performance of these methods. This omission points to a significant blind spot in assessing the overall robustness of the proposed frameworks.

Response: The discussion is added to address the potential impact of leverage points on the proposed methods.

11- Comment: The assertion that the techniques are limited to scenarios where the number of observations exceeds the number of predictors is an important limitation that could be explored further. A brief discussion on the implications of applying these methods in high-dimensional settings could clarify the next research steps.

Response: To address this, we have expanded the discussion to explore the implications of this limitation and identified future research directions to adapt the methods for high-dimensional settings.

12- Comment: In the spirit of reproducible research, the authors should provide both the data and the codes. These can be deposited on GitHub, for example.

Response: Both the data and the codes are uploaded on GitHub.

---

## [Decision Letter · Decision Letter 1]

3 Mar 2025

Principal fitted component framework for robust support vector regression based on bounded loss: A simulation study with potential applications

PONE-D-24-60709R1

Dear Dr. Tahir,

We’re pleased to inform you that your manuscript has been judged scientifically suitable for publication and will be formally accepted for publication once it meets all outstanding technical requirements.

Kind regards,

Mohamed R. Abonazel, Ph.D.

Academic Editor

PLOS ONE

Reviewers' comments:

Reviewer's Responses to Questions

**Comments to the Author**

1. If the authors have adequately addressed your comments raised in a previous round of review and you feel that this manuscript is now acceptable for publication, you may indicate that here to bypass the “Comments to the Author” section, enter your conflict of interest statement in the “Confidential to Editor” section, and submit your "Accept" recommendation.

Reviewer #1: All comments have been addressed

Reviewer #2: All comments have been addressed

2. Is the manuscript technically sound, and do the data support the conclusions?

Reviewer #1: (No Response)

Reviewer #2: Yes

3. Has the statistical analysis been performed appropriately and rigorously? 

Reviewer #1: (No Response)

Reviewer #2: I Don't Know

4. Have the authors made all data underlying the findings in their manuscript fully available?

Reviewer #1: (No Response)

Reviewer #2: Yes

5. Is the manuscript presented in an intelligible fashion and written in standard English?

Reviewer #1: (No Response)

Reviewer #2: Yes

6. Review Comments to the Author

Reviewer #1: (No Response)

Reviewer #2: Authors have revised the article. There were sugesstion to improe the article that were implemented.

7. PLOS authors have the option to publish the peer review history of their article (what does this mean? ). If published, this will include your full peer review and any attached files.

**Do you want your identity to be public for this peer review?** For information about this choice, including consent withdrawal, please see our Privacy Policy .

Reviewer #1: No

Reviewer #2: No

---

## [Editor Report · Acceptance letter]

PONE-D-24-60709R1

PLOS ONE

Dear Dr. Tahir,

I'm pleased to inform you that your manuscript has been deemed suitable for publication in PLOS ONE. Congratulations! Your manuscript is now being handed over to our production team.

Kind regards,

on behalf of

Dr Mohamed R. Abonazel

Academic Editor

PLOS ONE